# Robust Entropy Search for Safe Efficient Bayesian Optimization

**Dorina Weichert**[1] **Alexander Kister**[2] **Sebastian Houben**[3] **Patrick Link**[4,5] **Gunar Ernis**[1]

[1]Fraunhofer Institute for Intelligent Analysis and Information Systems IAIS, Sankt Augustin, Germany
[2]VP.1 eScience, Federal Institute for Materials Research and Testing BAM, Berlin, Germany
[3]University of Applied Sciences Bonn-Rhein-Sieg, Sankt Augustin, Germany
[4]Fraunhofer Institute for Machine Tools and Forming Technology IWU, Chemnitz, Germany
[5]Institute of Mechatronic Engineering, TUD Dresden University of Technology, Dresden, Germany

## Abstract

The practical use of Bayesian Optimization (BO) in engineering applications imposes special requirements: high sampling efficiency on the one hand and finding a robust solution on the other hand. We address the case of adversarial robustness, where all parameters are controllable during the optimization process, but a subset of them is uncontrollable or even adversely perturbed at the time of application. To this end, we develop an efficient information-based acquisition function that we call Robust Entropy Search (RES). We empirically demonstrate its benefits in experiments on synthetic and real-life data. The results show that RES reliably finds robust optima, outperforming state-of-the-art algorithms.

## 1 INTRODUCTION

**Motivation** Bayesian Optimization (BO) is a method for optimizing black-box functions that are costly to evaluate. It is used in various application domains, such as chemistry, robotics, or engineering [Shields et al., 2021, Berkenkamp et al., 2023, Lam et al., 2018]. The BO framework consists of three ingredients: i) a Bayesian surrogate model of the unknown black-box function, traditionally a Gaussian Process (GP) regression model, ii) an acquisition function to specify the next evaluation point based on the surrogate model, and iii) the evaluation process of the black-box function. Two fundamental properties that motivate the practical usage of BO are a high sample efficiency (i.e., a fast convergence regarding the number of function evaluations) and robustness against noisy evaluations of the underlying black-box function [Garnett, 2023, Shahriari et al., 2016].

The sample efficiency of BO depends heavily on the choice of the acquisition function. One class of acquisition functions are information-theoretic approaches, such as Entropy Search (ES) [Hennig and Schuler, 2012], Predictive Entropy Search [Hernández-Lobato et al., 2014], Max Value Entropy Search (MES) [Wang and Jegelka, 2017], Joint Entropy Search (JES) [Hvarfner et al., 2022], and $H_{l,A}$-Entropy Search [Neiswanger et al., 2022]. In all variations, the following evaluation point is chosen such that it maximizes the information gain about the (unknown) global optimum. This line of reasoning is more sample-efficient than that of other acquisition functions, such as Expected Improvement (EI) [Jones et al., 1998], Knowledge Gradient (KG) [Frazier et al., 2008] or Upper Confidence Bounds (UCB)-based [Srinivas et al., 2010] approaches but comes with higher computational cost [Garnett, 2023].

While BO is intrinsically robust against observation noise, as it is included into the surrogate model [Shahriari et al., 2016, Garnett, 2023], engineering applications are often required to be adversarially robust. We face this requirement using a setting with two kinds of parameters: parameters $x$ that are controllable during the optimization process and at application time (*controllable parameters*) and parameters $\theta$ that are controllable during the optimization process but externally affected at application time (*uncontrollable parameters*). A practical example of the latter set of parameters are environmental parameters, such as temperature, air pressure, or humidity, which are controllable in the lab but not at application time. An adversarially robust solution solves the following objective function:

$$x^\star, \theta^\star = \arg\min_x \arg\max_\theta f(x, \theta) \ . \tag{1}$$

It is an optimum of $f$, which is minimal even under maximal negative perturbation by the uncontrollable parameter $\theta$.

We are the first to tackle this problem with a sample-efficient information-theoretic acquisition function, Robust Entropy Search (RES). Closest to our work are the approaches of Bogunovic et al. [2018], who solve it by a UCB-based approach, and of Fröhlich et al. [2020], who treat the related problem of mean-case robustness against input noise by an information-theoretic approach.

**Contributions** Our contributions can be summarized as follows: First, we formulate the conditions for an optimum being an adversarially robust one and integrate them into the heart of the acquisition function - the probability distribution over the function values conditioned on these requirements. Subsequently, we delineate a step-by-step approach for practically applying this intermediate result within an acquisition function. Lastly, we provide a rigorous empirical evaluation of our approach, utilizing synthetic data and real-world scenarios from robotics and engineering.

## 2   RELATED WORK

Over the years, the traditional BO setting for pure minimization (see, e.g., [Shahriari et al., 2016, Garnett, 2023] for overviews) was enhanced to match several robustness requirements.

Prevalent is the treatment of input perturbations, i.e., input uncertainty, via a mean measure [Fröhlich et al., 2020, Beland and Nair, 2017, Nogueira et al., 2016, Iwazaki et al., 2021, Oliveira et al., 2019, Qing et al., 2022, Toscano-Palmerin and Frazier, 2018, 2022]: here, the objective is to minimize the expected value of an objective when the controllable parameters are perturbed, so to find $x^\star = \arg\min_x \mathbb{E}_{\theta \propto p(\theta)}[f(x + \theta)]$. As a result, these approaches are more likely to find a broad instead of a narrow optimum.

In our work, we instead investigate a more conservative case: adversarially robust optimization that finds a worst-case optimal solution $x^\star = \arg\min_x \max_\theta f(x, \theta)$. Bogunovic et al. [2018] treated this case a special case of their groundbreaking StableOpt algorithm that relies on the UCB approach by Srinivas et al. [2010]. Superficially, adversarially robust optimization was also treated by Weichert and Kister [2020] who adopt Thompson Sampling, ES and KG for discrete $\theta$. Recently, Christianson and Gramacy [2023] introduced an adversarially robust version of EI, dealing with a worst-case perturbation of the input, thus searching for the special case $x^\star = \arg\min_x \max_\theta f(x + \theta)$. In our approach, adding the input parameters is just one possible special case.

A further extension of the adversarially robust problem setting is distributionally robust optimization, where the goal is to find an optimum that is robust to a distributional shift within an uncertainty set $U$ of an uncontrollable parameter: $x^\star = \arg\min_x \sup_{Q \in U} \mathbb{E}_{\theta \propto Q}[f(x, \theta)]$. The work of Kirschner et al. [2020] was the first approach to this problem utilizing UCB until Husain et al. [2022], Tay et al. [2022], Yang et al. [2023] developed further approaches. Although these methods are related, they are not in the scope of our work.

Only a few of the named approaches arise from the information-based acquisition functions. There are the method by Fröhlich et al. [2020] to treat input perturbations and the one by Weichert and Kister [2020] to treat

adversarially robust entropy search for uncontrollable parameters from a discrete space. Our contribution extends the existing research with an information-based adversarially robust acquisition function.

## 3   BACKGROUND

Before we delve deeper into the derivation of the acquisition function, we would like to revisit GPs and briefly explain some basic properties of the adversarially robust optimum.

### 3.1   GAUSSIAN PROCESS REGRESSION

GP regression is a non-parametric method to model an unknown function $f(z) : \mathcal{Z} \mapsto \mathbb{R}$ by a distribution over functions. The GP prior is defined such that any subset of function values is normally distributed with mean $\mu_0(z)$ and covariance $k(z, z')$ for any $z, z' \in \mathcal{Z}$ (w.l.o.g. we assume $\mu_0(z) = 0$ [Rasmussen and Williams, 2006]). Conditioning the prior on actual data $D_t = \{(z_1, y_1), \ldots, (z_t, y_t)\}$, where $y = f(z) + \epsilon$, $\epsilon \sim \mathcal{N}(0, \sigma_n)$, the predictive posterior distribution $p(f) \sim GP(m_t, v_t | D_t)$ is given by

$$
\begin{aligned}
m_t(z|D_t) &= \boldsymbol{k}(z)^T \boldsymbol{K}^{-1} \boldsymbol{y} \\
v_t(z|D_t) &= k(z, z) - \boldsymbol{k}(z)^T \boldsymbol{K}^{-1} \boldsymbol{k}(z) ,
\end{aligned}
\tag{2}
$$

with $[\boldsymbol{k}(z)]_i = k(z, z_i)$, $\boldsymbol{K}_{i,j} = k(z_i, z_j) + \delta_{ij}\sigma_n^2$, where $\delta_{ij}$ is the Kronecker delta, and $[\boldsymbol{y}]_i = y_i$.

GPs are common surrogate models in BO. Since we consider a set of controllable parameters $x \in \mathcal{X} = \mathbb{R}^{d_c}$ and a set of uncontrollable parameters $\theta \in \Theta = \mathbb{R}^{d_u}$, $z$ in the previous definitions is replaced by the concatenation of $x$ and $\theta$, in our case $\mathcal{Z} = \mathcal{X} \times \Theta$.

### 3.2   PROPERTIES OF THE ROBUST OPTIMUM

The robust optimum $(x^\star, \theta^\star)$ has to fulfill two nested conditions:

(a) **Its function value is maximal in the direction of the uncontrollable parameters $\theta$,** generating a maximizing function $g(x) = \max_\theta f(x, \theta)$ and an argmax function $\boldsymbol{h}(x) = \arg\max_\theta f(x, \theta)$.

(b) **The optimum minimizes the maximizing function** $g(x)$. In consequence, the robust minimum is generally neither the global maximum nor minimum, but there generally exist function values of $f$ that are smaller and function values of $f$ that are larger than the robust optimum.

The difference between the optima is visualized as an example in figure 1. Besides of the global robust optimum ($\blacklozenge$), we show the global maximum ($\blacktriangleright$), the global minimum ($\triangleleft$) and

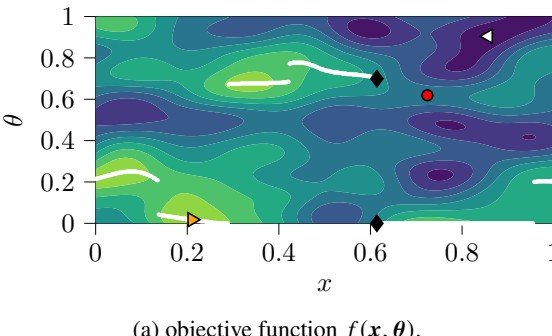

(a) objective function $f(x, \theta)$.

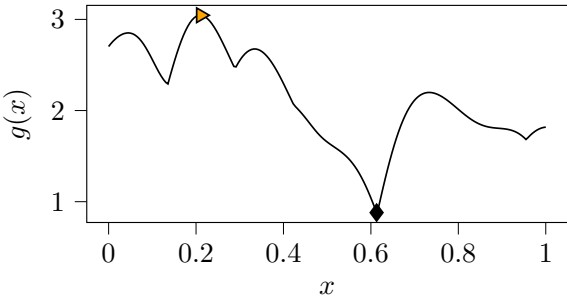

(b) maximizing function $g(x)$, derived from $f(x, \theta)$.

Figure 1: Two-dimensional objective function $f(x, \theta)$ and derived maximizing function $g(x) = \max_\theta f(x, \theta)$. In the given example, the location of the global robust optimum ($\blacklozenge$) is ambiguous. The optima are neither the global maximum ($\blacktriangleright$), the global minimum ($\triangleleft$) nor the smallest local min max point ($\bullet$). The values of the argmax function $h(x)$ are rendered as a white line in figure 1a. The function values at these points define the maximizing function $g(x)$, given in figure 1b.

the smallest local min max point ($\bullet$) (Nash equilibrium). Neither of the latter optima corresponds to the robust optimum that is sought.

## 4 ROBUST ENTROPY SEARCH

We propose the RES acquisition function that considers the properties of the robust optimum by involving the noiseless robust optimal value $f^\star = f(x^\star, \theta^\star)$, the argmax function $h(x)$ and its corresponding function values $g(x)$. Throughout the section we call these three quantities, $(h, g, f^\star)_f$ that all depend on $f$, *robustness characteristics*.

### 4.1 METHODICAL IDEA

Like other information-based acquisition functions, see, e.g., MES [Wang and Jegelka, 2017] or JES [Hvarfner et al., 2022], RES deduces the optimum by means of mutual information $I$ between the value $y(z) = f(z) + \varepsilon$ at the proposed location $z$ and some property of the optimum, in our case, the robustness characteristics $(h, g, f^\star)_f$. RES follows

$$
\begin{aligned}
\alpha_{RES}(z) &= I\left((z, y), (h, g, f^\star)_f \mid D_t\right) \\
&= H\left[p\left(y(z) \mid D_t\right)\right] \\
&\quad - \mathbb{E}_{(h,g,f^\star)_f}\left[H\left[p\left(y(z) \mid (h, g, f^\star)_f, D_t\right)\right]\right] \\
&\approx H\left[p\left(y(z) \mid D_t\right)\right] \\
&\quad - \frac{1}{C}\sum_{f_c \in \mathcal{F}_c} H\left[p\left(y(z) \mid (h_c, g_c, f_c^\star)_{f_c}, D_t\right)\right],
\end{aligned}
\tag{3}
$$

where $\mathcal{F}_c$ is a set of $C$ functions sampled from the actual GP posterior $GP(m_t, v_t \mid D_t)$ for the purpose of approximation. For each individual sample $f_c \in \mathcal{F}_c$, we find the corresponding robustness characteristics $(h_c, g_c, f_c^\star)_{f_c}$. As these

quantities follow a joint distribution, only one expectation is taken.

As we not only involve $f^\star$ but also the argmax function $h(x)$ and the maximizing function $g(x)$, the acquisition function proposes points that are likely to reduce the uncertainty about all robustness characteristics simultaneously.

The approximation of the conditional distribution $p(y(z) \mid (h_c, g_c, f_c^\star)_{f_c}, D_t)$ lies at the center of the acquisition function. In a first step, we simplify it by approximating noisy $y$ with $f$, since the observation noise is additive and can be added later when computing the entropy. Secondly, we implement the conditions formulated in section 3.2 into the conditional distribution. Therefore, we use indicator functions denoted by $\mathbb{1}_{\{\cdot\}}$:

$$
\begin{aligned}
p&\left(f \mid (h_c, g_c, f_c^\star)_{f_c}, D_t,\right) \\
&\propto \int df \, p(f \mid D_t) \cdot \mathbb{1}_{\{f(x, \theta) \le g_c(x)\}} \\
&\quad \cdot \mathbb{1}_{\{f_c^\star \le f(x, h_c(x)) \le g_c(x)\}}
\end{aligned}
\tag{4}
$$

The first indicator function implements the requirement of the optimum to be the maximum over the uncontrollable parameters $\theta$, referring to condition (a). By the second indicator function, we aim to find the minimum of these maxima by using the sampled optimum $f_c^\star$ as a lower bound on the distribution of maximum function values, implementing condition (b). Equation (4) is already a simplification and approximation of the actual target in equation (3): Instead of conditioning on the whole extreme functions $h$ and $g$, we only condition on the values of these functions at $(x, \theta)$.

### 4.2 IMPLEMENTATION

Our approach relies on the efficient treatment of samples from a GP and on the efficient calculation of the posterior predictive distribution, conditioned on the robustness charac-

teristics. We summarize all necessary implementation steps in the following.

### 4.2.1 Efficient Treatment of Function Samples

To efficiently sample from the actual GP, we make use of the Sparse Spectrum Gaussian Process (SSGP) approximation by Lázaro-Gredilla et al. [2010], which offers the opportunity to draw GP samples that have a closed analytical expression. This is beneficial for our approach, as we have to find the robustness characteristics numerically. Samples formed by this GP approximation are effectively optimized using gradient descent methods as derivatives are also available.

The function samples are of the form $f_c(z) = \boldsymbol{a}^T \boldsymbol{\phi}(z)$, with weight vector $\boldsymbol{a}$ and a vector of feature functions $\boldsymbol{\phi}(z) \in \mathbb{R}^F$, where $F$ is the number of feature functions. The elements $i$ of the feature vector $\boldsymbol{\phi}$ are given by $\phi_i(z) = \cos\left(\boldsymbol{w}_i^T z + b_i\right)$ with $b_i \sim U(0, 2\pi)$ and $\boldsymbol{w}_i \sim p(\boldsymbol{w}) \propto s(\boldsymbol{w})$ where $s(\boldsymbol{w})$ is the Fourier dual of the covariance function $k$. The elements of the weight vector $\boldsymbol{a}$ follow a normal distribution $\mathcal{N}\left(A^{-1}\boldsymbol{\Phi}^T y, \sigma_n^2 A^{-1}\right)$, with $A = \boldsymbol{\Phi}^T\boldsymbol{\Phi} + \sigma_n^2 I$, $\boldsymbol{\Phi}^T$ being the matrix composed from the feature function evaluated at the input data $\boldsymbol{\Phi}^T = [\boldsymbol{\phi}(z_1), \ldots, \boldsymbol{\phi}(z_t)]$, and $y$ being the corresponding observed function values. The sampling of functions therefore takes place in two steps: First, we draw frequencies $\boldsymbol{w}_i$ and phases $b_i$ to generate an unbiased approximation of the covariance function [Rahimi and Recht, 2007]. We then draw as many weight vectors $\boldsymbol{a}$ as function samples are needed from the resulting normal distribution. The resulting function samples can be evaluated cost-effectively by simple matrix-vector multiplication. For more details, see, e.g. the work of Lázaro-Gredilla et al. [2010] or Hernández-Lobato et al. [2014].

To find the argmax function $\boldsymbol{h}_c(x)$ and maximum value $g_c(x)$, a standard numerical solver, e.g. a gradient descent method, is called on-the-fly. To find the robust optimum, we implement a nested numerical solver that calculates the actual maximum over the uncontrollable parameters at every minimization step over the controllable ones.

### 4.2.2 Calculating the Conditioned Posterior Probability Distribution

A key ingredient for RES is the calculation of the conditional probability $p(f \,|\, (\boldsymbol{h}_c, g_c, f_c^\star)_{f_c}, D_t)$ in equation (4). As directly working on the function space is complex, we take a three-step approach to approximate the conditioned posterior probability distribution, inspired by the ideas of Fröhlich et al. [2020] and Hoffman and Ghahramani [2015]. Our final approximation is normal-distributed, and we can leverage the fact that the entropy of a normal distribution is given analytically for calculating the acquisition function.

**Step 1: Conditioning the GP at the training data points.** Instead of taking into account the whole GP on $\mathcal{X} \times \Theta$, we consider it only on a discrete subset of points from $\mathcal{X} \times \Theta$: the already evaluated training data points $D_t$. We enforce equation (4) to be true for all $z_i = (x_i, \theta_i) \in D_t$. Therefore, after calculating the maximizing uncontrollable parameters $\boldsymbol{h}_c(x_i)$ and their corresponding function values $g_c(x_i) = f_c(x_i, \boldsymbol{h}_c(x_i))$, we condition $\boldsymbol{f} = [f(z_1), \ldots, f(z_t), f(x_1, \boldsymbol{h}_c(x_1)), \ldots, f(x_t, \boldsymbol{h}_c(x_t))]^T$ on the robustness characteristics by Expectation Propagation (EP) [Minka, 2001]:

$$p\left(\boldsymbol{f} \,\middle|\, (\boldsymbol{h}_c(x), g_c(x), f_c^\star)_{f_c}, D_t\right)$$

$$\propto p\left(\boldsymbol{f} | D_t\right) \prod_{i=1}^{t} \mathbb{1}_{\{f(x_i, \theta_i) \le g_c(x_i)\}}$$

$$\cdot \mathbb{1}_{\{f_c^\star \le f(x_i, \boldsymbol{h}_c(x_i)) \le g_c(x_i)\}} \qquad (5)$$

$$\overset{\text{(EP)}}{\approx:} \mathcal{N}(\boldsymbol{\mu}_1, \boldsymbol{\Sigma}_1).$$

We approximate by EP, because problems of the form of equation (5) can only be solved analytically for lower dimensions [Rosenbaum, 1961, Ang and Chen, 2002]. EP has been shown to efficiently approximate the required measures in a reasonable computation time [Fröhlich et al., 2020, Gessner et al., 2020, Hennig and Schuler, 2012]. We reuse the implementation for linearly constrained Gaussians by Fröhlich et al. [2020], building on the work of Herbrich [2005] and reformulate the indicator functions to lower bounds $\boldsymbol{l}_b = \left[\boldsymbol{0}^{(1 \times t)}, f_c^{\star(1 \times t)}\right]^T$ and upper bounds $\boldsymbol{u}_b = [g_c(x_1), \ldots, g_c(x_t), g_c(x_1), \ldots, g_c(x_t)]^T$ to find the approximation $\mathcal{N}(\boldsymbol{\mu}_1, \boldsymbol{\Sigma}_1)$.

**Step 2: Creating a predictive distribution for a new location $z$.** We obtain a predictive distribution by marginalizing over the function values $\boldsymbol{f}$, using GP arithmetic. Already looking ahead to step three, we predict at $\hat{z} = [(x, \theta), (x, \boldsymbol{h}_c(x))]^T$, receiving predictions $p(f(\hat{z})|D_t, \boldsymbol{f}) = \mathcal{N}(\boldsymbol{\mu}_f, \boldsymbol{\Sigma}_f)$. We find

$$p_0(f(\hat{z}) \,|\, (\boldsymbol{h}_c(x), g_c(x), f_c^\star)_{f_c}, D_t)$$

$$= \int p(\boldsymbol{f} \,|\, (\boldsymbol{h}_c(x), g_c(x), f_c^\star)_{f_c}, D_t)$$

$$\cdot p(f(\hat{z})|D_t, \boldsymbol{f}) \mathrm{d}\boldsymbol{f} \qquad (6)$$

$$\approx \int \mathcal{N}(\boldsymbol{\mu}_1, \boldsymbol{\Sigma}_1) \cdot \mathcal{N}(\boldsymbol{\mu}_f, \boldsymbol{\Sigma}_f) \, \mathrm{d}\boldsymbol{f}$$

$$\approx \mathcal{N}(f(\hat{z})|m_0(\hat{z}), v_0(\hat{z})).$$

The predictive distribution again follows a normal distribution.

**Step 3: Conditioning the predictions.** As we only required the robustness conditions to be true for the training

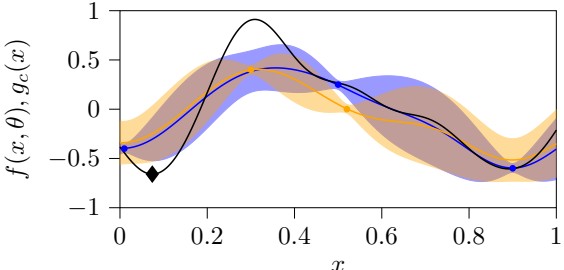

(a) predictive distribution and sample before conditioning

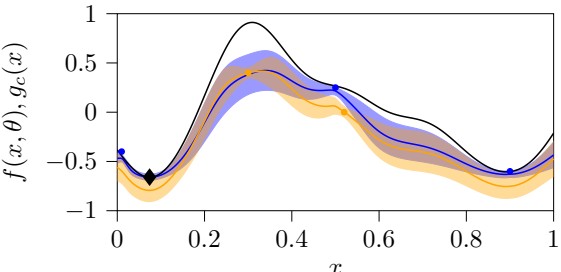

(b) predictive distribution after conditioning on the sample $g_c(x)$ and the robust sample optimum $f_c^\star$

Figure 2: Predictive distributions (mean and one standard deviation) before and after conditioning for a single uncontrollable parameter with two values $\theta_1$ (blue) and $\theta_2$ (orange). In this case, $\boldsymbol{h}_c(\boldsymbol{x}) = \theta_1 \,\forall\, \boldsymbol{x}$ (blue) with the max function $f_c(\boldsymbol{x}, \theta_1) = g_c(\boldsymbol{x})$ (black). While the predictive distribution $f(\boldsymbol{x}, \theta_2)$ is only upper bounded by the sample $g_c(\boldsymbol{x})$, $f(\boldsymbol{x}, \theta_1)$ is upper bounded by the sample $g_c(\boldsymbol{x})$ and lower bounded by the optimum $f_c^\star$ ($\blacklozenge$).

set $D_t$ in step 1, we now apply them to the predictions:

$$
\begin{aligned}
p(f(\hat{z})| & \left(\boldsymbol{h}_c(\boldsymbol{x}), g_c(\boldsymbol{x}), f_c^\star\right)_{f_c}, D_t) \\
&= \mathcal{N}(f(\hat{z})|m_0(\hat{z}), v_0(\hat{z})) \cdot \mathbb{1}_{\{f(\boldsymbol{x},\boldsymbol{\theta}) \leq g_c(\boldsymbol{x})\}} \\
&\qquad\qquad \cdot \mathbb{1}_{\{f_c^\star \leq f(\boldsymbol{x}, \boldsymbol{h}_c(\boldsymbol{x})) \leq g_c(\boldsymbol{x})\}} \\
&\approx \mathcal{N}(f(\hat{z})|\hat{m}_q(\hat{z}), \hat{v}_q(\hat{z}))
\end{aligned}
\tag{7}
$$

For a single $z$, we find a bivariate doubly truncated Gaussian with bounds as in step 1, for which the matching first and second moments are known analytically [Ang and Chen, 2002] and given in appendix A. From the matching moments, we extract the ones corresponding to the original $z$ by indexing: $m_q(z) = \hat{m}_{q(0)}, v_q(z) = \hat{v}_{q(0,0)}$.

In figure 2, we show the effect of conditioning for a problem with one discrete uncontrollable parameter $\boldsymbol{\theta} = \{\theta_1, \theta_2\}$ with two possible values. The worst-case function sample $g_c(\boldsymbol{x})$ originates from the blue uncontrollable parameter value $\theta_1$. The resulting posterior predictive distribution is changed as follows: on the one hand, all function values are upper-bounded by the maximizing function sample; on the other hand, the function values of $f(\boldsymbol{x}, \theta_1)$ are additionally lower-bounded by the sampled optimal value $f_c^\star$.

### 4.2.3 Final formulation of the RES acquisition function.

Given the final approximation $(m_q(z), v_q(z))$, we formulate the RES acquisition function as

$$
\begin{aligned}
\alpha_{\text{RES}}(z) &= \frac{1}{2} \log\left((v_t(z)|D_t) + \sigma_n^2\right) \\
&\quad - \frac{1}{2C} \\
&\quad \cdot \sum_{f_c \in \mathcal{F}_c} \log\left(\left(v_q(z)\,|\,\left(\boldsymbol{h}_c(\boldsymbol{x}), g_c(\boldsymbol{x}), f_c^\star\right)_{f_c}, D_t\right) + \sigma_n^2\right).
\end{aligned}
\tag{8}
$$

We summarize all necessary optimization steps in algorithms 1 and 2. In each iteration $t$, an SSGP approximation of the actual GP is calculated, and $C$ function samples are drawn. The robust optima $f_c^\star$ are calculated for these samples. Then, the GP is conditioned on the resulting robustness characteristics at the actual training data points. This step is only performed once when optimizing the acquisition function. Afterward, creation of the predictive distribution and conditioning at the new point $z$ is performed individually for each $z$ that is called during optimization of the acquisition function. Finally, we return the robust optimum of the actual model's predictive mean $m_t$.

In appendix C.1.1, we provide results on the time complexity of our algorithm for different combinations of discrete and continuous variables. Overall, the runtime is governed by the calculation of equation (6), with effects from calculating the argmax function $\boldsymbol{h}_c(\boldsymbol{x})$ and the GP prediction to obtain $\mathcal{N}(\boldsymbol{\mu}_f, \boldsymbol{\Sigma}_f)$.

---

**Algorithm 1** Robust BO with RES acquisition function.

---

**Input** maximum number of iterations $T$, space of controllable parameters $\mathcal{X}$, space of uncontrollable parameters $\Theta$, number of samples $C$, size of initial design $M$

**Output** robust optimum $z^\star = (x^\star, \theta^\star)$

1: $D_M \leftarrow \{z_i, y_i\}_{i=1}^M$
2: **for** $t = M, \dots, M + T - 1$ **do**
3: $\quad GP(m_t(z), v_t(z)) \leftarrow \textsc{FitGP}(D_t)$
4: $\quad \mathcal{F}_c \leftarrow \textsc{SampleGP}(GP, C)$ ▷ Create SSGP, sample
5: $\quad \mathcal{F}_c^\star \leftarrow \emptyset$
6: $\quad$ **for** $c = 1, \dots, C$ **do**
7: $\qquad \mathcal{F}_c^\star \leftarrow \mathcal{F}_c^\star \cup f_c^\star = \min_{x \in \mathcal{X}} \max_{\theta \in \Theta} f_c(\boldsymbol{x}, \boldsymbol{\theta})$
8: $\quad$ **end for**
9: $\quad z_{t+1} \leftarrow \arg\max_{z \in \mathcal{X} \times \Theta} \alpha_{RES}(z, GP, \mathcal{F}_c, \mathcal{F}_c^\star)$
10: $\quad y_{t+1} = f(z_{t+1}) + \epsilon, D_{t+1} \leftarrow D_t \cup \{z_{t+1}, y_{t+1}\}$
11: **end for**
12: **return** $(x^\star, \theta^\star) \leftarrow \arg\min_{x \in \mathcal{X}} \arg\max_{\theta \in \Theta} m_t(\boldsymbol{x}, \boldsymbol{\theta})$

---

**Algorithm 2** The RES acquisition function.

**Input** evaluation point $z$, GP $GP$, function samples $\mathcal{F}_c$, robust optima $\mathcal{F}_c^\star$
**Output** value of RES acquisition function
1: $H \leftarrow 0$
2: **for** $c \in \{1, \ldots, C\}$ **do**
3:      **if** IsNotInitialized($\alpha_{RES}$) **then**
4:          $\mu_1, \Sigma_1 \leftarrow$ ApproximateEP($GP, f_c, f_c^\star$)
5:          $\triangleright$ sec. 3.2.2., step 1
6:      **end if**
7:      $h_c(x) \leftarrow \arg\max_{\theta \in \Theta} f_c(x, \theta)$
8:      $g_c(x) \leftarrow f_c(x, h_c(x))$
9:      $v_q(z) \leftarrow$
10: ConditionPosteriorVariance($\mu_1, \Sigma_1, h_c, g_c$)
11:          $\triangleright$ sec. 3.2.2, steps 2 & 3
12:      $H \leftarrow H + \log(v_q(z) + \sigma_n^2)$
13: **end for**
14: **return** $\alpha_{RES} \leftarrow \frac{1}{2}\log(v_t(z) + \sigma_n^2) - \frac{1}{2C}H$

# 5 EXPERIMENTS

We conduct three types of experiments: in a preliminary test, we estimate the general performance of the acquisition function and its dependency on the number of necessary function samples $C$ in a within-model comparison. Secondly, we compare our algorithm with state-of-the-art benchmarks on synthetic problems. Finally, two real-life problems are treated: the calibration of parameters of a numerical simulation, arising in an engineering task, and robust robot pushing, an experiment formulated by Bogunovic et al. [2018].

We compare our approach, RES, to StableOpt [Bogunovic et al., 2018] with different exploration constants $\sqrt{\beta}$, and with the non-robust acquisition functions MES [Wang and Jegelka, 2017], UCB [Srinivas et al., 2010], KG [Frazier et al., 2008], and with standard EI [Jones et al., 1998]. In RES, we set the number of features $F = 500$ for the SSGP. For MES, we choose a value of a number of 100 sampled minima, and the exploration parameter $\sqrt{\beta}$ in UCB was set to a value of 2. For KG, which was originally designed for discrete spaces, we discretize the continuous space of dimensionality $d_{\text{conti.}}$ by a random grid of size $50^{d_{\text{conti.}}}$ drawn from a uniform distribution in each iteration and use a number of 32 function samples. For StableOpt, based on the experiments in the original publication, we apply constant exploration constants from $\sqrt{\beta} \in \{1, 2, 4\}$.

For measuring performance, we use algorithm- and problem-specific metrics. As RES evaluates at a location that raises the knowledge about the optimum and not at a potential optimum location, the optimum location is calculated at every iteration as the robust optimum of the actual model mean $z_t^\star = (x^\star, \theta^\star) = \arg\min_{x \in \mathcal{X}} \arg\max_{\theta \in \Theta} m_t(x, \theta)$. The other approaches evaluate locations that might be the optimum; for them we assume $z_t^\star = (x^\star, \theta^\star) = \arg\max_{z \in \mathcal{X} \times \Theta} \alpha(z|D_t)$.

Given these optima, we calculate regret measures. For problems with a discrete space of uncontrollable parameters $\Theta$, where $h(x)$ is cheap to calculate, we directly take into account our robustness requirement by evaluating the robust regret $|f(x^\star, h(x^\star)) - f^\star|$. For problems with uncontrollable parameters from a continuous space $\Theta$, such as the within-model comparison, $h(x)$ is hardly accessible. Therefore, we use the inference/immediate regret $|f(z_t^\star) - f^\star|$ for the evaluation of the RES acquisition function/the other acquisition functions. Notably, the metrics are non-monotonic, as the guess about the optimum can deteriorate with time. However, using a monotonic measure like best regrets, i.e., specifying the regret of the best found optimum up to iteration $t$ for each run, is not helpful for min max problems. This is because pure minimization algorithms can find an optimum close to the robust optimum at the beginning of the optimization process and then converge to a non-robust optimum. The use of best regrets obscures this behavior.

If not mentioned otherwise, we use a zero mean GP prior, and a squared-exponential covariance function with automatic-relevance detection $k(z, z') = \sigma_v^2 \exp\left(-0.5\|z - z'\|^2_{L^{-1}}\right)$ with $L = \text{diag}\left[l_{c1}^2, \ldots, l_{d_c}^2, l_{u1}^2, \ldots, l_{d_u}^2\right]$.

Runtime results for representative experiments are given in appendix C.1.2.

The code to conduct the experiments is built on open source implementations of GPs [GPy, since 2012], BO [Paleyes et al., 2023], SSGPs and EP [Fröhlich et al., 2020] and publicly available at https://github.com/fraunhofer-iais/Robust-Entropy-Search.

## 5.1 WITHIN-MODEL COMPARISON

For the within-model comparison, we follow the approach of Hennig and Schuler [2012] to compare the acquisition functions independently from the correct fit of the actual GP model.

Therefore, we use a GP model with squared-exponential covariance function with signal variance $\sigma_f^2 = 1$, a constant lengthscale of $l = 0.1$ in all dimensions and a noise variance of $\sigma_n^2 = 0.001$. For each of the 50 initializations, we draw 1000 random data points whose locations follow a uniform distribution in $[0, 1]^2$ and whose values are distributed according to a normal distribution with zero mean and the covariance according to the specified covariance function. Given these points, we initialize a GP. Its predictive mean is employed as the objective function for optimization, so we deal with a two-dimensional continuous problem. The motivating example in figure 1 is one of the resulting optimization problems. For RES, we apply numbers of function samples of $C \in \{1, 5, 10, 30\}$.

In figure 3, we report the results of the experiments. As

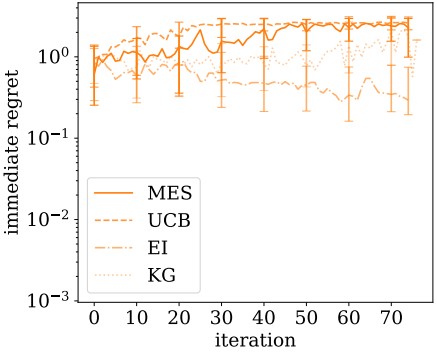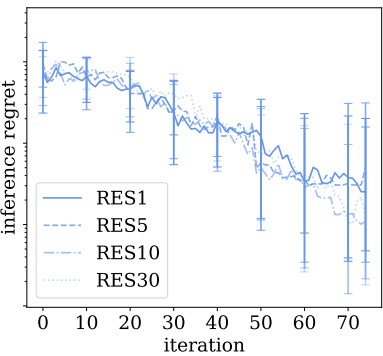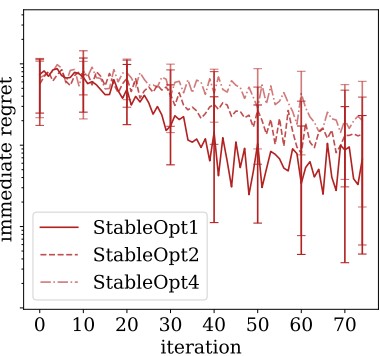

Figure 3: Regrets for the two-dimensional, continuous within-model comparison. We present the median and the upper and lower quartiles for 50 GP mean functions. The number after the algorithm indicates the value of the hyperparameter ($C$ for RES and $\sqrt{\beta}$ for StableOpt). The results indicate the failure of the non-robust methods as well as the fact that RES acquisition function is slightly better than StableOpt with the advantage of being hyperparameter-free.

expected, the non-robust approaches are not able to find the robust optima. For StableOpt, the performance on this particular problem depends on the value of the exploration parameter - (for the within-model comparison) the lower, the better. Generally, our approach RES is better than StableOpt and, advantageously, does not require setting a hyperparameter. Additionally, the number of samples only slightly impacts the performance of RES. Therefore, we set the number of samples to 1 for all other experiments.

## 5.2 SYNTHETIC BENCHMARK FUNCTIONS

In the synthetic experiments, we measure the performance of our approaches on problems with unknown hyperparameters. Basically, we use variations of the Branin [Surjanovic and Bingham, 2013], the Sinus + Linear [Fröhlich et al., 2020], the Eggholder [Surjanovic and Bingham, 2013], the Hartmann 3D [Surjanovic and Bingham, 2013], and the Synthetic Polynomial [Bertsimas et al., 2010] functions. In these originally non-robust optimization problems, we declare a subset of dimensions as uncontrollable parameters $\theta$ and then search for the robust optimum.

To reduce the computational effort, we discretize the space of the uncontrollable parameters $\Theta$ in all experiments. The exact number of uncontrollable parameters is given below the figures, as well as the problem's dimensionality. Full details and visualizations of the individual problems are given in appendix B.

We run each algorithm with 50 initializations. For all problems, except from the Synthetic Polynomial where we fix the hyperparameters, we optimize the hyperparameters of the GP model in every iteration via maximum likelihood. The noise hyperparameter $\sigma_n$ is fixed to a value of 0.001 in all problems.

In figure 4, we report the performance of all algorithms in terms of the quartiles. The experiments show a superior per-

formance of RES over the other approaches, independent from the dimensionality or the complexity of the problem (e.g., the eggholder problem having a lot of local optima). In some cases, algorithms oscillate between different optima, i.e., for the StableOpt algorithm with $\sqrt{\beta} = 4$ in the Sinus + Linear problem and for the non-robust algorithms in the Synthetic Polynomial. This is due to the very different values of the max function $g(x)$ for different inputs $x$. Additionally, the previously en par StableOpt algorithm struggles with the fixed or unknown hyperparameters of the model. This behavior was already reported for plain UCB in Hennig and Schuler [2012] and seems to apply also for the robust adaption. Also, due to the unknown hyperparameters, StableOpt underlies the risk of too early exploitation. In these cases, one of the local robust optima is preferred over the global one, increasing the width of the distribution over results. Therefore, StableOpt often reaches better results in the lower quantiles (if it examines the correct local optimum). However, its median behavior is worse than that of our approach, as RES is forced to explore more globally as it has to learn not only about the robust optimum but also about the other robustness characteristics. This behavior becomes particularly clear in the Sinus + Linear, the Eggholder and the Hartmann problems.

In appendix C.1.3 we additionally provide results on the robust regret over the runtime for the Branin function. RES achieves a similar regret in the same time as StableOpt with a significantly lower number of iterations.

## 5.3 REAL-LIFE BENCHMARK PROBLEMS

We treat two benchmark problems connected to applying robust BO in real life.

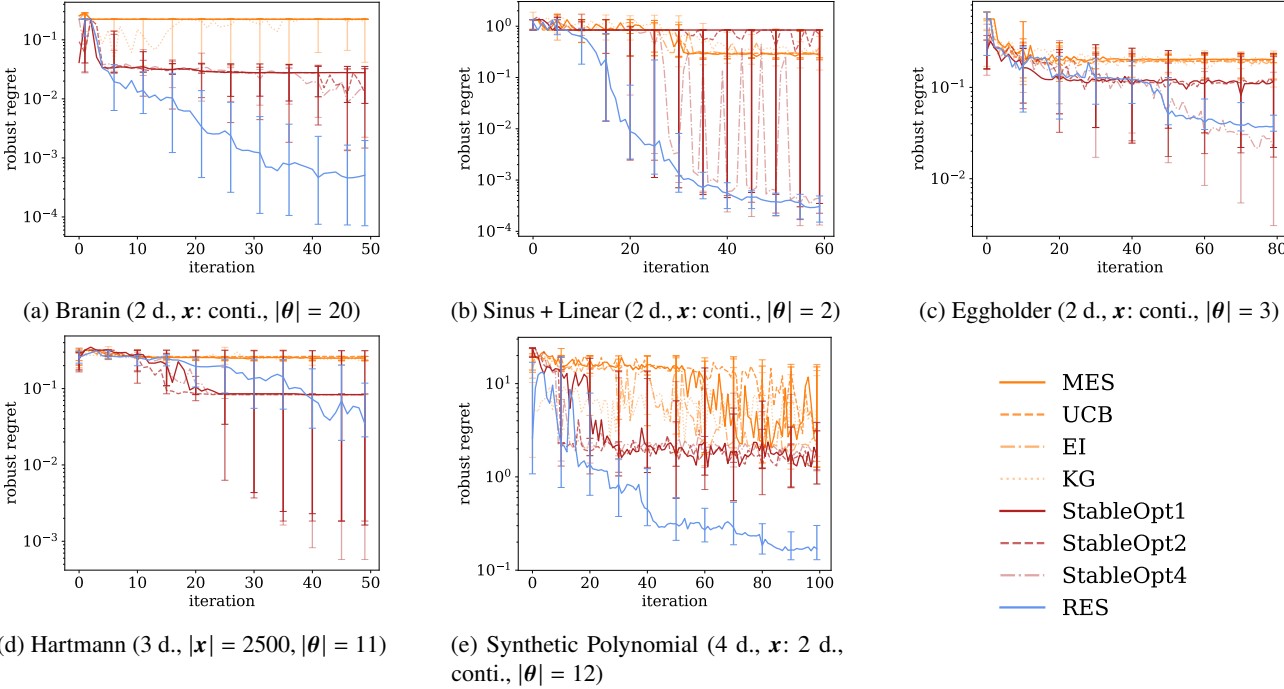

(a) Branin (2 d., $x$: conti., $|\theta| = 20$)

(b) Sinus + Linear (2 d., $x$: conti., $|\theta| = 2$)

(c) Eggholder (2 d., $x$: conti., $|\theta| = 3$)

(d) Hartmann (3 d., $|x| = 2500$, $|\theta| = 11$)

(e) Synthetic Polynomial (4 d., $x$: 2 d., conti., $|\theta| = 12$)

Figure 4: Results of the experiments with synthetic functions. The marker after the name of the problem indicates the dimensionality, the type of input space (continuous or discrete) and, if discrete, the number of discrete parameters. For StableOpt, we give the value of the exploration constant $\sqrt{\beta}$ after the algorithm name. Our approach, RES, with a number samples of $C = 1$, shows superior performance on nearly all problems.

#### 5.3.1 Calibration of Finite Element Method Simulation Parameters

In engineering disciplines, a lot of research and development tasks involve the application of heavy simulations, e.g., simulations via Finite Element Method. These simulations are typically taking minutes to months for execution. Nevertheless, they are advantageous over real-life experiments in the lab, as they are often cheaper (they do not induce, e.g., material costs) and enable insights of multiple metrics at each time step and many locations simultaneously. Unfortunately, the approximation quality of simulations depends on material parameters, which are typically unknown. These parameters are not directly measurable and depend on exogenous parameters, such as local temperature. Therefore, a set of experiments is taken out at a range of different uncontrollable parameters, and engineers use the result to calibrate the simulations, i.e., to fit the unknown (controllable) material parameters to approximate the experiments.

In our use case, we treat the calibration of simulation parameters of a deep drawing process, where one simulation takes about 12 minutes on 16 cores of an Intel(R) Core(TM) i9-10980XE processor. In deep drawing, a metal sheet is placed on a die, held in place by a blank holder, and drawn into a new shape by pressing a punch, see figure 5a. Experimentally, the force of the punch $F_{\mathrm{punch}_{\mathrm{ex}}}$ was measured over time, varying the constant force of the blank-holder

$F_{\mathrm{holder}} \in \{200, 300, 350\}$ kN. The static coefficient of friction $\mu_H \in [0.1, 0.2]$ is treated as the controllable parameter, which depends, as no lubrication is used, only on the (unknown) surface quality of the die, punch, blank holder, and the metal sheet. Exemplary experimental and simulated force-time diagrams are shown in figure 5b. The optimization objective is to minimize the maximum absolute difference between the experimental and simulated punch force, so we seek to find $\mu_H^\star = \arg\min_{\mu_H} \max_{F_{\mathrm{Holder}}} |F_{\mathrm{punch}_{\mathrm{ex}}} - F_{\mathrm{punch}_{\mathrm{sim}}}| = \arg\min_{\mu_H} \max_{F_{\mathrm{holder}}} f(\mu_H, F_{\mathrm{Holder}})$.

We run our optimization approach 30 times for 25 iterations for the RES and the EI acquisition function, each with one random sample for initialization. Hyperparameters of the model are estimated in every iteration via maximum likelihood. To find the robust optimum for comparison, we join the data from all 750 evaluations, create a GP model, and calculate the function value at the model's optimum. Figure 5c shows the optimization results in terms of robust regret: while EI soon finds some non-robust optimum, RES finds a considerably better robust optimum already after ten iterations. While more iterations would have been interesting from a scientific perspective, the results were already sufficient for the application side. The robust optimal coefficient of friction $\mu_H^\star$ is now used as a safe estimate for simulations with unknown blank-holder force.

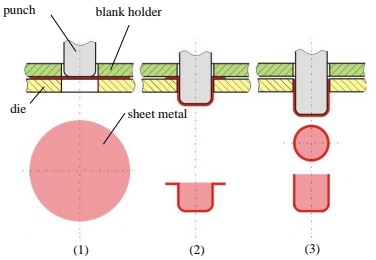

(a) Deep Drawing Process [Ivo, 2007]

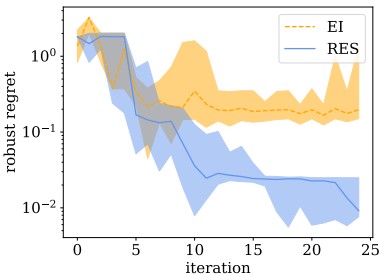

(b) Simulated and experimental results of Force-Time Diagram for $\mu_H = 0.2$.

(c) Robust Regret for RES and EI acquisition functions.

Figure 5: Deep drawing: schematic illustration, force-time-diagrams and regret curves for simulations with different parameters.

### 5.3.2 Robust Robot Pushing

In appendix C.2, we provide results on the robust robot pushing problem from Bogunovic et al. [2018] and find RES again performing best.

## 6 CONCLUSION

We introduced a novel worst-case robust acquisition function for BO, RES. In a nutshell, this acquisition function simultaneously maximizes the information gain about the robust objective function $g$, the location of the robust objective function $\boldsymbol{h}$, and the robust optimal value $f^\star$. In several benchmark experiments, we demonstrate the superior efficiency of our acquisition function and show its benefit in two use cases from engineering and robotics.

## 7 LIMITATIONS AND FUTURE WORK

This paper's main contribution is developing an innovative information-theoretic acquisition function for adversarially robust BO. When used with a sufficiently accurate model, it produces impressive results. However, its performance relies on the correctness of the model, which is not necessarily the case for complex problems. A simple technique to detect a poor model fit is via the $\gamma$-exploit approach, used by Hvarfner et al. [2022], where in each iteration, with probability $\gamma$, the actual optimum is evaluated. Unfortunately, this approach detects but does not circumvent a poor model fit. Therefore, we expect an even more significant improvement in combination with automatic model selection methods, such as those by Malkomes and Garnett [2018], Gardner et al. [2017]. Especially the ability to discover additive structures in the work of Gardner et al. [2017] promises to additionally scale the approach to higher-dimensional spaces, thus being a valuable enhancement.

Additionally, the derivation of regret bounds would likewise be interesting, such as typically performed in UCB-based approaches, such as the StableOpt algorithm [Bogunovic

et al., 2018]. For information-based acquisition functions, we are only aware of the disputed [Takeno et al., 2022] regret bounds for MES and its descendants [Wang and Jegelka, 2017, Belakaria et al., 2019]. An extension of the existing work, considering the recent discussions and the robust setting of our approach, is a challenging open problem.

For future work, we intend to adapt our algorithm to various domains, such as the constrained [Gelbart et al., 2014, Gardner et al., 2014], the multi-fidelity [Forrester et al., 2007], and multi-objective [Swersky et al., 2013] setting.

### Author Contributions

D. Weichert conceived the idea of the paper together with A. Kister, created the code, the figures, wrote the initial draft of the manuscript and performed revisions. A. Kister conceived the idea of the paper and revised the manuscript. P. Link performed the simulations via Finite Element Method. S. Houben revised the manuscript. G. Ernis revised the initial draft of the manuscript.

### Acknowledgements

We thank the reviewers for their helpful feedback. The work of D. Weichert has been funded by the Federal Ministry of Education and Research of Germany and the state of North Rhine-Westphalia as part of the Lamarr Institute for Machine Learning and Artificial Intelligence, Sankt Augustin, Germany. P. Link was funded by the Deutsche Forschungsgesellschaft (DFG, German Research Foundation) - 438646126.

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

# Adversarially Robust Entropy Search for Safe Efficient Bayesian Optimization

**Dorina Weichert**[1] **Alexander Kister**[2] **Sebastian Houben**[3] **Patrick Link**[4,5] **Gunar Ernis**[1]

[1]Fraunhofer Institute for Intelligent Analysis and Information Systems IAIS, Sankt Augustin, Germany
[2]VP.1 eScience, Federal Institute for Materials Research and Testing BAM, Berlin, Germany
[3]University of Applied Sciences Bonn-Rhein-Sieg, Sankt Augustin, Germany
[4]Fraunhofer Institute for Machine Tools and Forming Technology IWU, Chemnitz, Germany
[5]Institute of Mechatronic Engineering, TUD Dresden University of Technology, Dresden, Germany

## A  APPROXIMATION OF BIVARIATE DOUBLY TRUNCATED GAUSSIAN

We closely follow the results of Ang and Chen [2002]. Let $\boldsymbol{x} = (x_1, x_2) \sim \mathcal{N}(\boldsymbol{0}, \boldsymbol{\Sigma})$, with lower bounds $\boldsymbol{l}_b = \begin{bmatrix} l_{b_1} & l_{b_2} \end{bmatrix}^T$ and upper bounds $\boldsymbol{u}_b = \begin{bmatrix} u_{b_1} & u_{b_2} \end{bmatrix}^T$, and $\rho$ denote the correlation of the two variables.

Let the cumulative density be denoted by $L$

$$L(\boldsymbol{l}_b, \boldsymbol{u}_b) = \int_{l_{b_1}}^{u_{b_1}} \int_{l_{b_2}}^{u_{b_2}} f_{\boldsymbol{x}}(x_1, x_2) dx_1 dx_2 \, ,$$

with $f_{\boldsymbol{x}}(x_1, x_2)$ being the density function of $\boldsymbol{x}$. $L$ can be evaluated numerically, e.g., using the method of Genz [1992].

For the moments, we find:

$$m_{10} = \frac{1}{L} \left[ \psi(l_{b_1}, u_{b_1}, l_{b_2}, u_{b_2}) + \rho \psi(l_{b_1}, u_{b_1}, l_{b_2}, u_{b_2}) \right] \tag{9}$$

$$m_{20} = \frac{1}{L} \left[ L + \chi(l_{b_2}, u_{b_2}, l_{b_1}) - \chi(l_{b_2}, u_{b_2}, u_{b_1}) + \rho^2 \chi(l_{b_1}, u_{b_1}, l_{b_2}) - \rho^2 \chi(l_{b_1}, u_{b_1}, u_{b_2}) \right] \tag{10}$$

$$\begin{aligned} m_{11} = \frac{1}{L} \Big[ & \rho L + \rho \Upsilon(l_{b_1}, u_{b_1}, l_{b_2}) - \rho \Upsilon(l_{b_1}, u_{b_1}, u_{b_2}) + \rho \Upsilon(l_{b_2}, u_{b_2}, l_{b_1}) - \rho \Upsilon(l_{b_2}, u_{b_2}, u_{b_1}) \\ & + \Lambda(l_{b_1}, u_{b_1}, l_{b_2}) - \Lambda(l_{b_1}, u_{b_1}, u_{b_2}) \Big] \end{aligned} \tag{11}$$

with helper functions

$$\psi(l_{b_1}, u_{b_1}, l_{b_2}, u_{b_2}) = \phi(l_{b_1}) \left[ \Phi\left( \frac{u_{b_2} - \rho l_{b_1}}{\sqrt{1-\rho^2}} \right) - \Phi\left( \frac{l_{b_2} - \rho l_{b_1}}{\sqrt{1-\rho^2}} \right) \right] - \phi(u_{b_1}) \left[ \Phi\left( \frac{u_{b_2} - \rho u_{b_1}}{\sqrt{1-\rho^2}} \right) - \Phi\left( \frac{l_{b_2} - \rho u_{b_1}}{\sqrt{1-\rho^2}} \right) \right] \, ,$$

$$\begin{aligned} \chi(l_{b_2}, u_{b_2}, l_{b_1}) = \; & l_{b_1} \phi(l_{b_1}) \left[ \Phi\left( \frac{u_{b_2} - \rho l_{b_1}}{\sqrt{1-\rho^2}} \right) - \Phi\left( \frac{l_{b_2} - \rho l_{b_1}}{\sqrt{1-\rho^2}} \right) \right] \\ & + \frac{\rho \sqrt{1-\rho^2}}{\sqrt{2\pi} \left( 1+\rho^2 \right)} \left[ \phi\left( \frac{\sqrt{l_{b_2}^2 - 2\rho l_{b_2} l_{b_1} + l_{b_1}^2}}{\sqrt{1-\rho^2}} \right) - \phi\left( \frac{\sqrt{u_{b_2}^2 - 2\rho u_{b_2} l_{b_1} + l_{b_1}^2}}{\sqrt{1-\rho^2}} \right) \right] \, , \end{aligned}$$

$$\Upsilon(l_{b_2}, u_{b_2}, l_{b_1}) = l_{b_1} \phi(l_{b_1}) \left[ \Phi\left( \frac{u_{b_2} - \rho l_{b_1}}{\sqrt{1-\rho^2}} \right) - \Phi\left( \frac{l_{b_2} - \rho l_{b_1}}{\sqrt{1-\rho^2}} \right) \right] \, ,$$

and

$$\Lambda(l_{b_2}, u_{b_2}, l_{b_1}) = \frac{\sqrt{1-\rho^2}}{\sqrt{2\pi}} \left[ \phi\left( \frac{\sqrt{l_{b_2}^2 - 2\rho l_{b_2} l_{b_1} + l_{b_1}^2}}{\sqrt{1-\rho^2}} \right) - \phi\left( \frac{\sqrt{u_{b_2}^2 - 2\rho u_{b_2} l_{b_1} + l_{b_1}^2}}{\sqrt{1-\rho^2}} \right) \right],$$

where $\phi$ is the probability density function and $\Phi$ is the cumulative density function of the standard normal $\mathcal{N}(0,1)$.

The moments $m_{01}$ and $m_{02}$ are obtained by interchanging $(l_{b_1}, u_{b_1})$ and $(l_{b_2}, u_{b_2})$ in the formulae for $m_{10}$ and $m_{20}$.

Given these moments, we finally find the following approximating normal distribution $\mathcal{N}(\hat{\boldsymbol{\mu}}, \hat{\boldsymbol{\Sigma}})$ with $\hat{\boldsymbol{\mu}} = \begin{bmatrix} m_{10} & m_{01} \end{bmatrix}^T$ and $\hat{\boldsymbol{\Sigma}} = \begin{bmatrix} m_{20} - m_{10}^2 & m_{11} - m_{10}m_{01} \\ m_{11} - m_{10}m_{01} & m_{02} - m_{01}^2 \end{bmatrix}$. From these, we extract $m_q = m_{10}$ and $v_q = m_{20} - m_{10}^2$.

## B   DETAILED DESCRIPTION OF EXPERIMENTS WITH SYNTHETIC BENCHMARK FUNCTIONS

**Branin Function**   The branin function is defined by

$$f(\boldsymbol{x}, \boldsymbol{\theta}) = a(\boldsymbol{\theta} - b\boldsymbol{x}^2 + c\boldsymbol{x} - r)^2 + s(1-t)\cos(\boldsymbol{x}) + s,$$

with $a = 1$, $b = 5.1/(4\pi^2)$, $c = 5/\pi$, $r = 6$, $s = 10$, and $t = 1/(8\pi)$ and is defined on $x \in [-5, 10]$, $\theta \in [0, 15]$ [Surjanovic and Bingham, 2013].

We use discrete values of the uncontrollable parameter $\theta \in \{0.75, 1, \ldots, 14, 14.25\}$, $|\Theta| = 20$, and scale the input space to $[0, 1]^2$ and the output values to $\mathcal{N}(0, 1)$.

For optimization, the hyperparameters of the GP are bounded to $\sigma_v \in [10^{-5}, 10]$ and $l \in [10^{-5}, 10]^2$. The model is initialized with a single random point from the domain. We run each algorithm with 50 different initializations for 50 iterations.

Figure 6a shows the original optimization problem with the 20 discrete values of the uncontrollable parameter as white horizontal lines, the robust optimum and the global minimum. The maximizing function $g(\boldsymbol{x}) = \max_{\boldsymbol{\theta}} f(\boldsymbol{x}, \boldsymbol{\theta})$ is visualized in figure 6b.

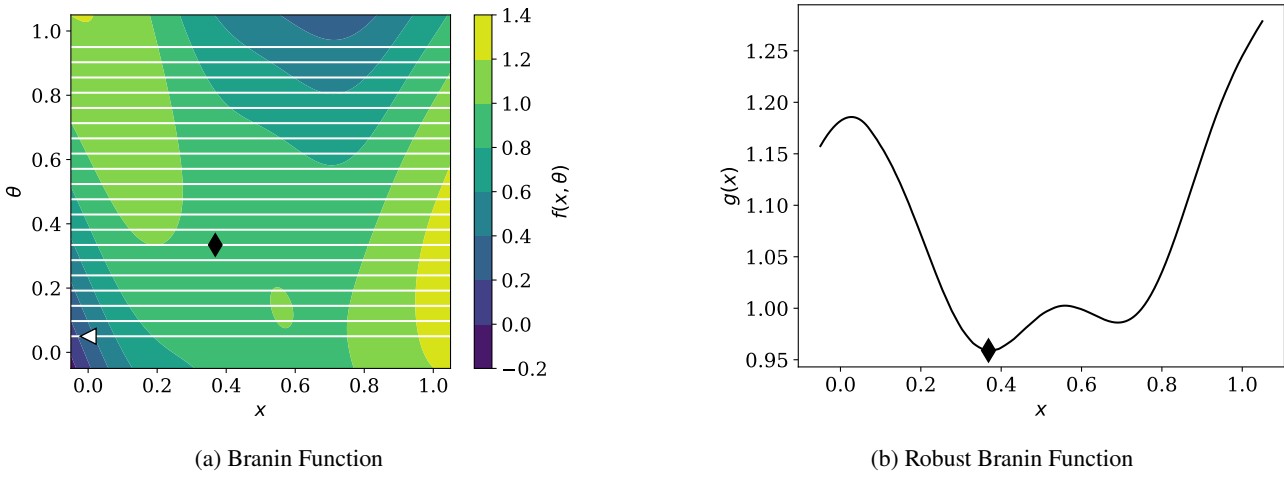

(a) Branin Function

(b) Robust Branin Function

Figure 6: Visualization of the robust variant of Branin Function. The global robust optimum is indicated by $\blacklozenge$, the global minimum by $\triangleleft$.

**Sinus + Linear Function**   The sinus + linear function is defined by

$$f(z) = \sin(5z^2\pi) + 0.5z,$$

where $z = x + \theta$ with $x \in [0, 1]$ and $\theta \in \{0.1, 0.05\}$. It was originally used by Fröhlich et al. [2020] with continuous $\theta \in [-0.05, 0.05]$. We opted for discretization for the sake of simplicity.

Figure 7a visualizes the problem. Multiple local robust and non-robust optima exist, which are close to each other.

For optimization, the hyperparameters of the GP are bounded to $\sigma_v \in \left[10^{-5}, 10\right]$ and $l \in \left[10^{-5}, 10\right]^2$. The model is initialized with a single random point from the domain. We run each algorithm with 50 different initializations for 60 iterations.

**Hartmann Function**  Following Surjanovic and Bingham [2013], the three-dimensional Hartmann function is defined by

$$f(z) = \sum_{i=1}^{4} \alpha_i \exp\left(-\sum_{j=1}^{3} A_{ij} \left(z_j - P_{ij}\right)^2\right),$$

where $\alpha = \begin{bmatrix} 1.0 & 1.2 & 3.0 & 3.2 \end{bmatrix}^T$, $A = \begin{bmatrix} 3 & 10 & 30 \\ 0.1 & 10 & 35 \\ 3 & 10 & 30 \\ 0.1 & 10 & 35 \end{bmatrix}$, $P = 10^{-4} \begin{bmatrix} 3689 & 1170 & 2673 \\ 4699 & 4387 & 7470 \\ 1091 & 8732 & 5547 \\ 381 & 5743 & 8828 \end{bmatrix}$. It is defined on $z \in [0,1]^3$. In

our experiments, we use the first two dimensions as controllable parameters $x$ on an equidistant grid of size $50 \times 50 = 2500$, and use the third dimension as uncontrollable parameter $\theta$, which is discretized to values of $\{0.25, 0.3, \ldots, 0.7, 0.75\}, |\Theta| = 11$.

The maximizing function $g(x) = \max_\theta f(x, \theta)$, the robust optimum and the global minimum are visualized in figure 7b.

For optimization, the hyperparameters of the GP are bounded to $\sigma_v \in \left[10^{-5}, 10\right]$ and $l \in \left[10^{-5}, 10\right]^2$. The model is initialized with a single random point from the domain. We run each algorithm with 100 different initializations for 50 iterations.

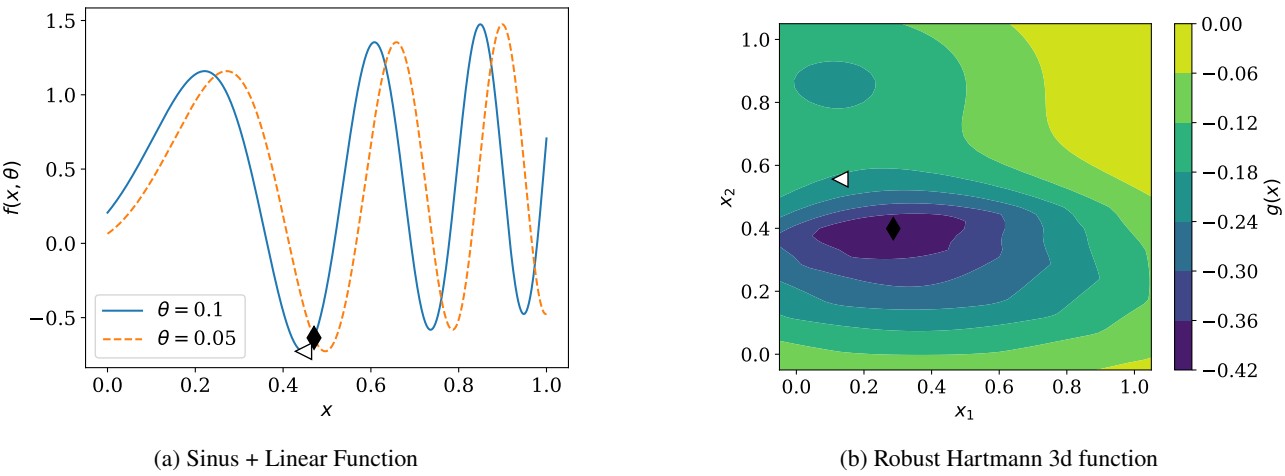

(a) Sinus + Linear Function

(b) Robust Hartmann 3d function

Figure 7: Visualization of robust Sinus + Linear and Hartmann function variants. The global robust optimum is indicated by ♦ and the global minimum by ◁.

**Eggholder Function**  Following Surjanovic and Bingham [2013], the eggholder function is defined by

$$f(x, \theta) = -(\theta + 47) \sin\left(\sqrt{\left|\theta + \frac{x}{2} + 47\right|}\right) - x \sin\left(\sqrt{|x - (\theta + 47)|}\right),$$

with $x \in [-512, 512]$, $\theta \in [-512, 512]$.

We use discrete values of the uncontrollable parameter $\theta \in \{-512, 0, 185\}$, and scale the input space to $[0, 1]^2$ and the output values to zero mean and a variance of 1.

Figure 8a shows the original optimization problem with the three uncontrollable parameters as white horizontal lines, as well as the robust optimum. The maximizing function $g(x) = \max_\theta f(x, \theta)$ is visualized in figure 8b.

For optimization, the hyperparameters of the GP are bounded to $\sigma_v \in \left[10^{-5}, 10\right]$ and $l \in \left[10^{-5}, 10\right]^2$. The model is initialized with a single random point from the domain. We run each algorithm with 50 different initializations for 80 iterations.

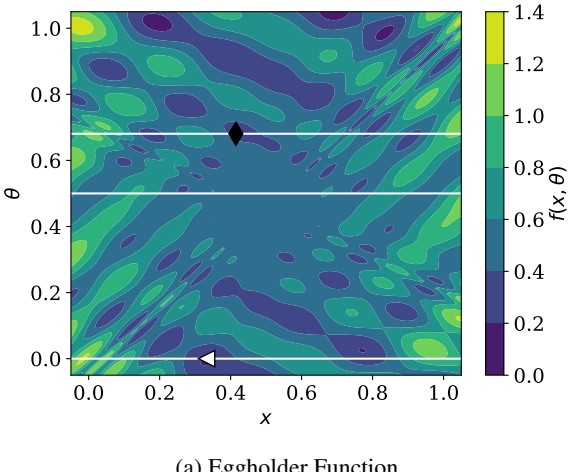 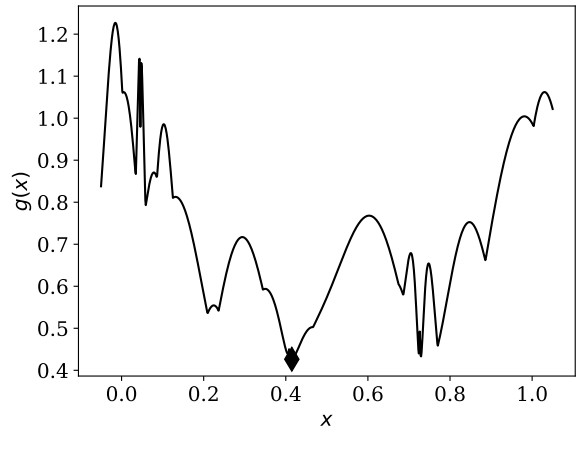

(a) Eggholder Function

(b) Robust Eggholder Function

Figure 8: Visualization of the robust variant of Eggholder Function. The global robust optimum is indicated by ♦, the global minimum by ◁.

**Synthetic Polynomial**   We adopt the synthetic polynomial, which has already been considered in multiple variations by Bertsimas et al. [2010], Bogunovic et al. [2018], Fröhlich et al. [2020], Christianson and Gramacy [2023]. It is originally defined by Bertsimas et al. [2010]:

$$f(z) = 2z_1^6 - 12.2z_1^5 + 21.2z_1^4 + 6.2z_1 - 6.4z_1^3 - 4.7z_1^2$$
$$+ z_2^6 - 11z_2^5 + 43.3z_2^4 - 10z_2 - 74.8z_2^3 + 56.9z_2^2$$
$$- 4.1z_1z_2 - 0.1z_2^2z_1^2 + 0.4z_2^2z_1 + 0.4z_1^2z_2$$

with $z = \begin{bmatrix} z_1 & z_2 \end{bmatrix}$. We choose $x_1 \in [-0.95, 3.2]$ and $x_2 \in [-0.45, 4.4]$, and $\theta$ in a circular neighborhood with radii $r \in \{0, 0.5\}$ and angles $\alpha \in \{0, 0.4\pi, 0.8\pi, 1.2\pi, 1.6\pi, 2\pi\}$, so $z = x + \theta = x + r \begin{bmatrix} \cos \alpha & \sin \alpha \end{bmatrix}$.

Figure 9a shows the original optimization problem ($\theta = 0$). The maximizing function $g(x) = \max_\theta f(x, \theta)$ is visualized in figure 9b. The robust optimum is far from the non-disturbed one.

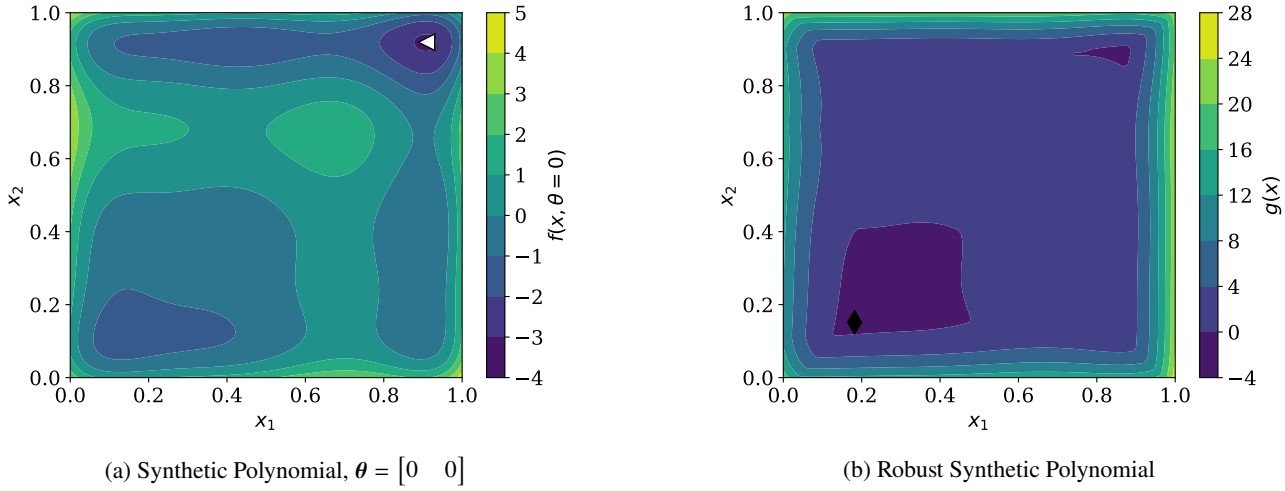

(a) Synthetic Polynomial, $\theta = \begin{bmatrix} 0 & 0 \end{bmatrix}$

(b) Robust Synthetic Polynomial

Figure 9: Visualization of the robust variant Synthetic Polynomial Problem. The global robust optimum is indicated by ♦, the global minimum by ◁.

Similar to Bogunovic et al. [2018], we fix the hyperparameters to values found by maximum likelihood estimation using 500 randomly sampled points with function values below 15. The model is initialized with ten random points from the domain. We run each algorithm with 100 different initializations for 100 iterations.

# C ADDITIONAL EXPERIMENT RESULTS

## C.1 RUNTIME RESULTS

### C.1.1 Time Complexity of Algorithm

To evaluate the time complexity of one iteration of RES, we have to consider the types of parameters, i.e. wether the (un)controllable parameters are discrete or continuous. Therefore, we distinguish four cases: the case of fully discrete parameters, the case of fully continuous parameters and the mixed ones.

For all of them, the calculation time is dominated by calculation of equation (6). Two aspects are influencing it: the marginalization step, which is of order $O\left(N^3\right)$, with $N$ being the number of data points in $D_t$, and the optimization procedure to find the argmax function $\boldsymbol{h}_c(\boldsymbol{x})$ and the corresponding function value $g_c(\boldsymbol{x})$, with a single prediction of the function sample $f_c$ from a set of $C$ function samples, each with $F$ Fourier feature functions scaling with $O\left(F^2\right)$.

An additional aspect to take into account is the scaling of the different applied optimization procedures. We apply the Nelder-Mead method, which (in the worst case of a nonconvex and nonsmooth function) scales with $O\left(\frac{d^2}{\psi^4}\right)$ to reach a required precision $\psi$ in dimensionality $d$ [Garmanjani and Vicente, 2013], the L-BFGS-B algorithm, which scales with maximum order $O(d)$ per iteration [Zhu et al., 1997], and simple maximization of $N_d$ data points, being of $O(N_d)$. Given these prerequisites, we can derive the complexity for all cases of parameter type combinations. In the following, we refer to the dimensionality of the uncontrollable parameters as $d_u$, to the dimensionality of the controllable parameters as $d_c$, to the number of uncontrollable parameters as $N_u$, and to the number of controllable parameters as $N_c$.

**Fully continuous parameters.** We search for the maximum of the acquisition function by multistart Nelder-Mead method with $N_R$ restarts. In each Nelder-Mead iteration, we have to call multistart the L-BFGS-B optimizer with $N_r$ restarts and $N_i$ iterations. Therefore, we find a complexity of $N_R O\left(\frac{(d_c+d_u)^2}{\psi^4}\right) C\left(O\left(N^3\right) + N_r N_i\left(O\left(d_u\right) + O\left(F^2\right)\right)\right)$.

**Fully discrete parameters.** In the fully discrete case, we have to evaluate all combinations of parameters and maximize afterward. For each controllable parameter, we have to find the maximizing value of the uncontrollable parameters. Therefore, we have to predict once and maximize $N_c$ times. Therefore, we find a complexity of $O(N_c N_u) + C\left(O\left(N^3\right) + O\left(F^2\right) + N_c O\left(N_u\right)\right)$.

**Continuous controllable parameters and discrete uncontrollable parameters.** In this case, we optimize the acquisition function again via multistart Nelder-Mead method but find the maximizing uncontrollable parameters in the discrete way. In each Nelder-Mead iteration, we have to maximize the function sample, and we find a complexity of $N_R O\left(\frac{d_c^2}{\psi^4}\right) C\left(O\left(N^3\right) + O\left(F^2\right) + O\left(N_u\right)\right)$.

**Discrete controllable parameters and continuous controllable parameters.** Even though we do not perform experiments for this case, we provide the result for sake of completeness. Here, the outer optimization is performed in a discrete manner, while the inner one is continuous, so the complexity scales with $O(N_c) + N_c\left(C\left(O\left(N^3\right) + N_r N_i\left(O\left(d_u\right) + O\left(F^2\right)\right)\right)\right)$.

### C.1.2 Practical Runtime Experiments

In tables 1, 2, and 3 we summarize the computation time of the algorithms for a fully continuous experiment (e.g., the within-model comparison), for a fully discretized experiment (e.g., the discretized Hartmann function), and an experiment that has a continuous space of controllable parameters $\mathcal{X}$ and a discrete space of uncontrollable parameters $\Theta$, (e.g., the Branin function). The measured runtime contains the initialization and the optimization of the acquisition function for one iteration. For StableOpt, we include the runtime for all values of the exploration constant $\sqrt{\beta}$. The experiments are taken out on Intel Xeon Gold 5118 CPUs, using 12 cores in parallel, for the Branin function, we were able to apply 24 cores.

Overall, the runtime of our approach RES is between StableOpt and KG, with KG being faster on the Branin function, which is due to it running on a small discrete space of controllable parameters $\mathcal{X}$ and RES on a continuous space of controllable parameters $\mathcal{X}$. We assume that the optimization of the RES acquisition function on the mixed space $\mathcal{X} \times \Theta$ takes more iterations than the optimization over its discrete version.

Table 1: Runtime results for within model comparison. Results in seconds. *: KG algorithm runs on a discretized space of $50 \times 50$.

| Quantile | RES | StableOpt | MES | KG* | UCB | EI |
|---|---|---|---|---|---|---|
| 25 % | 203.99 | 34.32 | 0.34 | 1197.04 | 0.12 | 0.17 |
| 50 % | 232.97 | 44.92 | 0.39 | 1525.47 | 0.15 | 0.22 |
| 75 % | 271.23 | 58.87 | 0.52 | 2043.63 | 0.24 | 0.39 |

Table 2: Runtime results for the fully discretized Hartmann function. Results in seconds.

| Quantile | RES | StableOpt | MES | KG | UCB | EI |
|---|---|---|---|---|---|---|
| 25 % | 195.413 | 0.021 | 0.117 | 1199.908 | 0.021 | 0.022 |
| 50 % | 199.438 | 0.026 | 0.125 | 2278.777 | 0.026 | 0.026 |
| 75 % | 204.000 | 0.031 | 0.133 | 3848.871 | 0.031 | 0.031 |

Table 3: Runtime results for the Branin function. Results in seconds. *: KG algorithm runs on a discretized space of $50 \times 1$.

| Quantile | RES | StableOpt | MES | KG* | UCB | EI |
|---|---|---|---|---|---|---|
| 25 % | 28.069 | 1.309 | 0.156 | 9.819 | 0.066 | 0.084 |
| 50 % | 33.536 | 3.593 | 0.167 | 12.373 | 0.072 | 0.093 |
| 75 % | 38.387 | 5.305 | 0.177 | 15.262 | 0.079 | 0.103 |

### C.1.3 Performance over Runtime

In figure 10, we provide the robust regret over the runtime for the Branin function. Even though RES experiences a slow start, it achieves a similar regret in the same time as StableOpt with a significantly lower number of iterations (as can be seen from the experiment in the main part of the paper).

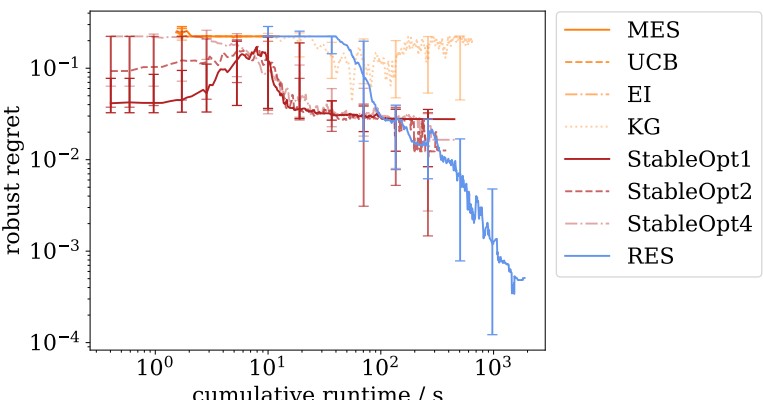

Figure 10: Regret over runtime for the Branin Problem. RES reaches the same robust regret as StableOpt in a similar amount of time.

### C.2 RESULTS FOR ROBUST ROBOT PUSHING PROBLEM

We adopt the robust robot pushing problem from Bogunovic et al. [2018], which is based on the publicly available code[1] of the robot pushing objective by Wang and Jegelka [2017].

---

[1]https://github.com/zi-w/Max-value-Entropy-Search

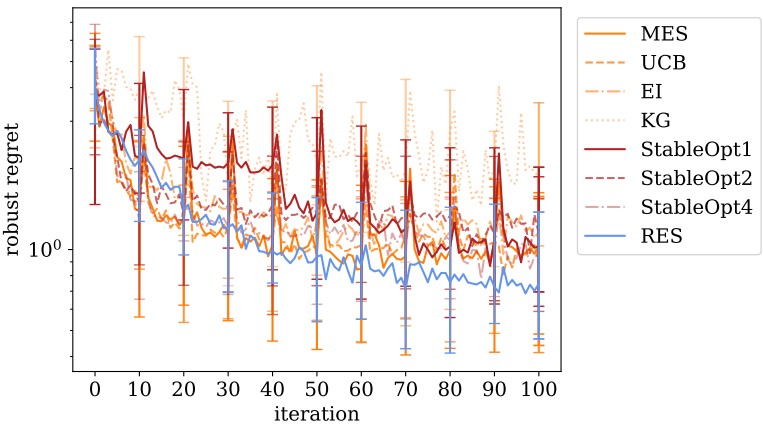

Figure 11: Results for robust robot pushing problem.

In the problem, a good pre-image for pushing an object to an unknown target location is sought. Precisely, there are two different target locations, where the first is uniformly distributed over the domain and the second uniform over the $l_1$-ball centered at the first target location with radius $r = 2.0$. Each evaluation calls a function $f(r_x, r_y, r_t) = 5 - d_{end}$, where $(r_x, r_y) \in [-5, 5]^2$ is the initial robot location, $r_t \in [1, 30]$ is the pushing duration and $d_{end}$ is the distance to the target location.

We run the problem 30 times for 100 iterations, where each initialization consists of a randomly drawn pair of targets and two starting positions, one for each target. We make a fully Bayesian treatment of the model hyperparameters, updated every 10th iteration. In figure 11, we report the robust regret: RES again shows a superior performance. The large discontinuities in the curves are caused by hyperparameter re-estimation.