# OpenReview forum: "Robust Entropy Search for Safe Efficient Bayesian Optimization"
_auai.org/UAI/2024/Conference — UAI 2024 poster_

### Official Review · Reviewer_pXLQ · 2024-03-05

**Q2-1 Originality-Novelty:** 2
**Q2-2 Correctness-Technical Quality:** 3
**Q2-5 Clarity Of Writing:** 3

**Q10 Ethical Concerns:**

No ethical concerns

**Q1 Summary And Contributions:**

This paper addresses the issue of the robust minimization of an objective black-box function $f$. More precisely, some parameters of $f$ (called _uncontrollable parameters_ and denoted $\mathbf \theta$) are assumed to be set in an adversarial fashion. The goal of a robust optimizer is to set the remaining parameters (called _controllable parameters_ and denoted $\mathbf x$) so that the function value $f(\mathbf x, \mathbf \theta)$ is minimized. In essence, this task is the minimax optimization of a black-box function.

To this end, the paper introduces a new acquisition function, called Robust Entropy Search (RES). After describing its theoretical motivation, the paper provides useful implementation details and illustrates RES' empirical performance.

**Q2-3 Extent To Which Claims Are Supported By Evidence:**

2: Fair: the main claims are somewhat supported by evidence (but the experimental evaluation may be weak, or does not match entirely with the claims, important baselines may be missing, proofs contain important ideas but lack rigor, algorithmic details are only discussed superficially, references are imprecise, assumptions are not sufficiently motivated or explicated, etc.).

**Q2-4 Reproducibility:**

2: Fair: key resources (e.g. proofs, code, data) are unavailable but key details (e.g. proof sketches, experimental setup) are sufficiently well-described for an expert to confidently reproduce the main results.

**Q3 Main Strengths:**

The authors seek to make BO safer by taking into account potential exogenous effects on some parameters of the objective function. This is an important problem that many industrial actors face, and it is adequately motivated in the paper.

RES appears sound and shows interesting empirical performance.

Overall the paper is well-written.

**Q4 Main Weakness:**

The weaknesses are labelled for future reference in the discussion period.

(W1) Although RES appears sound, it is not supported by a regret bound. I believe that a regret analysis would significantly strengthen the contribution.

(W2) Another main concern of mine is about runtime results of RES. According to Appendix C.1, runtime results are provided for the Within-Model experiment, the Hartmann function and the Branin function. RES appears to be 4x, 7000x (!) and 60x slower than StableOpt, respectively. I understand that RES may still be useful for applications where the objective function is very expensive to evaluate (timewise or moneywise). However, I think it is a major drawback since, for a vast majority of applications, a user might prefer using a faster algorithm able to collect more data (and hence, potentially achieving better results) in a given time budget.

(W3) I also wonder about the complexity of RES. Such an analysis would have given valuable insights about RES to the reader.

(W4) I believe the empirical evaluation is unfair to SafeOpt (see my questions in the next section).

**Q5 Detailed Comments To The Authors:**

I am interested in discussing with the authors about Weaknesses (W1) to (W4). Here are some related questions and suggestions to spark a discussion:

(1) Have you tried to derive a regret bound for RES?

(2) A relevant additional figure would be to plot the robust regret w.r.t. wall-clock time. Doing so would help figure out whether RES achieves better results even though it is unable to collect as much data as other robust algorithms such as SafeOpt.

(3) Do you have any insight about how well SafeOpt scales w.r.t. the dimensionality of the objective function domain? w.r.t. a desired accuracy $\xi$ on the robust minimizer?

(4) Lemma 1 in [1] provides an expression for the exploration parameter of SafeOpt, which is clearly a function of time. Why was SafeOpt evaluated with constant, arbitrary exploration parameters instead? Additionally, why is SafeOpt not evaluated on the calibration of FEM simulation parameters? How does RES compare to SafeOpt with its exploration properly set? What is the performance of SafeOpt on calibrating FEM simulation parameters?

[1] Ilija Bogunovic, Jonathan Scarlett, Stefanie Jegelka, and Volkan Cevher. Adversarially robust optimization with gaussian processes. In Advances in Neural Information Processing Systems, volume 31, 2018.

**Q9 Complying With Reviewing Instructions:**

Yes

---

> ### Author Rebuttal · Authors · 2024-04-04
>
> Thank you for your helpful comments.
>
> **(W1) + (1)**: Unfortunately, for most of the information-based acquisition functions, even for Entropy Search from 2012, no regret bounds or an approach how to derive them have been presented in literature so far to the best of our knowledge. Even though a regret bound would indeed strengthen our contribution, we had to follow the regret-bound-free tradition for RES.
>
> **(W2) + (2) and (W3) + (3)**: For the complexity of RES, please have a look at the results we provide to reviewer T2Hr, where we find the runtime of finding a new point scale by $O_{\text{cont.}}$, respectively $O_{\text{disc.}}$.
> If we now optimize the acquisition function, by e.g. Nelder-Mead approach with $N_{r_{\text{outer}}}$ restarts, which, under some assumptions, scales with $\mathcal{O}(\frac{d^2}{\varepsilon})$ to reach a required precision $\varepsilon$ [1], where $d$ is the dimensionality of the problem, we find the overall complexity for optimization of the acquisition function to scale with $N_{r_{\text{outer}}}(\mathcal{O}(\frac{(d_c+d_u)^2}{\varepsilon})O_{\text{cont.}})$ for the continuous case.
> In the case of discrete uncontrollable parameters, we have to evaluate the acquisition functions for all uncontrollable parameters and maximize, thus we find the algorithm to scale with $N_{r_{\text{outer}}}(\mathcal{O}(\frac{d_u^2}{\varepsilon^2})(\mathcal{O}(N_u)+N_uO_{\text{disc.}}))$.
> In the fully discrete case, we have to evaluate all points to find the next combination of $\mathbf{x}$ and $\boldsymbol{\theta}$ to acquire. Given $N_u$ uncontrollable and $N_c$ controllable parameters, we have to evaluate $N_uN_c$ times and find the maximum of these values, so we achieve a runtime of order $\mathcal{O}(N_cN_u) + N_cN_uO_{\text{disc.}}$.
>
> For StableOpt, we can use an analogous way to derive a complexity result.
> To find the next point for evaluation, StableOpt uses a two-step approach:
> 1. $\tilde{\mathbf{x}}\_t = \arg \min\_{x \in \mathcal{X}} \max\_{\boldsymbol{\theta} \in \Theta} \text{lcb}(\mathbf{x}, \boldsymbol{\theta})$.
> 2. $\boldsymbol{\theta}\_t = \arg \max\_{\boldsymbol{\theta} \in \Theta} \text{ucb}(\tilde{\mathbf{x}}\_t, \boldsymbol{\theta})$.
>
> The calculation of $\text{lcb}$ and $\text{ucb}$ requires a Gaussian Process prediction, scaling $\mathcal{O}(N^3)$.
>
> Let's treat the different cases of variable combinations:
> - continuous case: with multistart Nelder-Mead method and L-BFGS-B algorithm for the inner/outer optimization, we find a complexity of $N_{r_{\text{outer}}}\mathcal{O}(\frac{(d_u+d_c)^2}{\varepsilon^2})N_rN_i (\mathcal{O}(d_u)+\mathcal{O}(N^3))$ for the first step and $N_rN_i (\mathcal{O}(d_u)+\mathcal{O}(N^3))$ for the second step.
> - mixed variable types: as we predict for all uncontrollable parameters simulatenously and maximize afterward, we have to exchange the term $N_rN_i (\mathcal{O}(d_u)+\mathcal{O}(N^3))$ by $\mathcal{O}(N^3)+\mathcal{O}(N_u)$ for both steps.
> - fully discrete case: In the first step, we have to make a single prediction of $\mathcal{O}(N^3)$ and find its min max, involving the ordering of the uncontrollable parameters for each controllable one, being of scale $N_c\mathcal{O}(N_u)$ and than ordering the result, being an operation of $\mathcal{O}(N_c)$. We therefore find a combined complexity of $\mathcal{O}(N^3)+N_c\mathcal{O}(N_u)+\mathcal{O}(N_c)$. The additional second step is afterward to predict and then to maximize, thus scaling by $\mathcal{O}(N^3)+\mathcal{O}(N_u)$.
>
> Even though these results do not exactly your question about a scaling w.r.t a desired accuracy on the robust minimizer, it explains the runtime differences we report in our experiments. The cost of the inner optimization problem in RES exceeds the one in StableOpt.
>
> We are currently rerunning the Branin experiment to create the suggested figure and hope to provide ituntil the end of the rebuttal phase. Thank you for this suggestion.
>
> **(W4) + (4)**: Even though the mentioned Lemma 1 suggests a theoretically time-dependent value of the exploration parameter, the authors themselves state that they found it over-confident and therefore use a constant value of 2 in their experiments. Therefore, we also use constant values in our experiments.
>
> Regarding the simulation calibration, we are unfortunately not able to run these simulations again due to their runtime and license cost. We propose to instead rerun the experiment with StableOpt using a surrogate model based on all simulations taken so far. We suppose that this approach can answer your concern.
>
> **Closing remarks**: As already mentioned to the other reviewers, we will improve the paper, especially the technical part, until the end of rebuttal phase and for the camera-ready version.
> We hope to have sufficiently addressed your concerns and would be grateful for further discussions.
>
> [1] Konevcn'y, J., & Richtárik, P. (2014). Simple Complexity Analysis of Simplified Direct Search. arXiv: Optimization and Control.

---

### Official Review · Reviewer_5D9e · 2024-03-21

**Q2-1 Originality-Novelty:** 3
**Q2-2 Correctness-Technical Quality:** 3
**Q2-5 Clarity Of Writing:** 3

**Q1 Summary And Contributions:**

The authors introduce a novel worst-case robust acquisition function for BO, RES.

It |simultaneously maximizes the information gain about the robust objective function 𝑔, the location of the robust objective function h, and the robust optimal value 𝑓."

It is demonstrated successfully in both synthetic and real world benchmarks.

**Q2-3 Extent To Which Claims Are Supported By Evidence:**

2: Fair: the main claims are somewhat supported by evidence (but the experimental evaluation may be weak, or does not match entirely with the claims, important baselines may be missing, proofs contain important ideas but lack rigor, algorithmic details are only discussed superficially, references are imprecise, assumptions are not sufficiently motivated or explicated, etc.).

**Q2-4 Reproducibility:**

3: Good: key resources (e.g. proofs, code, data) are available and key details (e.g. proofs, experimental setup) are sufficiently well-described for competent researchers to confidently reproduce the main results.

**Q3 Main Strengths:**

They authors claim that "Generally, our approach RES is better than StableOpt and, advantageously, does not require setting a hyperparameter." This is indeed quite a strong results and the main strength of this contribution.

**Q4 Main Weakness:**

I’d appreciate a more detailed discussions of why the following happens: “Even though StableOpt often reaches better results in the lower quantiles, its median behavior is worse than our approach’s. Additionally, it has the risk of too early exploitation, as in the Branin, the Sinus + Linear, and the Hartmann 3d problems for the lower exploration constants.”

Why is it better in lower quantiles?
Why does it have that risk?
What’s the theoretical/intuitive reasoning beyond the given empirical observations?

**Q5 Detailed Comments To The Authors:**

a) Figure 3 can be improved, maybe just collapse it into one like int Fig 4?

**Q9 Complying With Reviewing Instructions:**

Yes

---

> ### Author Rebuttal · Authors · 2024-04-04
>
> Thank you for your helpful comments.
>
> In the following, we will discuss the mentioned behavior of the algorithms in more detail.
>
> Therefore, we first have to examine the behavior of \sopt. Here, the next point to be sampled depends on a hyperparameter-dependent weighting of the predictive mean $m_t(\mathbf{z})$ and predictive variance $v_t(\mathbf{z})$ of the Gaussian process, where the hyperparameter is the exploration constant $\beta^{^\frac{1}{2}}$ that we vary in the experiments. StableOpt finds the next point to be acquired by a two-stage process, making use of the upper and lower confidence bounds of the Gaussian Process prediction, $\text{ucb}(\mathbf{z}) = m_t(\mathbf{z}) +\beta^{^\frac{1}{2}} \sqrt{v_t(\mathbf{z})}$, $\text{lcb}(\mathbf{z}) = m_t(\mathbf{z}) - \beta^{^\frac{1}{2}} \sqrt{v_t(\mathbf{z})}$. In line with all UCB-based approaches, $\beta^{^\frac{1}{2}}$ governs the exploration behavior: If $\beta^{^\frac{1}{2}}$ is small, the acquisition function will tend to require data close to the best value that has been evaluated, if it is high, it will more explore the space $\mathcal{Z}$.
>
> The Gaussian Process prediction used in the acquisition function relies on the estimated hyperparameters $\boldsymbol{L}$ and $\sigma_v$ of the covariance function $k(\mathbf{z}, \mathbf{z}') = \sigma_v^2 \exp \left(-0.5 \vert\vert \mathbf{z} - \mathbf{z}' \vert\vert^2_{\mathbf{L}^{-1}}\right)$ with $\mathbf{L} = \text{diag}\left[l^2_{c1}, \dots, l^2_{d_c}, l^2_{u1}, \dots, l^2_{d_u} \right]$, that are estimated in every iteration of the Bayesian Optimization procedure by maximum likelihood method.
> Especially the dimension-wise entries in the lengthscale matrix $\mathbf{L}$  govern the assumed behavior of the underlying function. If they are small, the model assumes a wiggly underlying function in the respective dimension, if it is high, the function is assumed to be smooth.
> If now the exploration constant $\beta^{\frac{1}{2}}$ is too small, the acquisition function will generate a lot of data close to the first local robust optimum found. If this is the case, the model will become a self-fulfilling prophecy, as the fitted lengthscale will become only as small as necessary to sufficiently cover the data. Therefore, \sopt with a too small exploration constant risks too early exploitation, which goes hand in hand with a fast convergence to some optimum. If each run converges early, we gain a broad distribution over results for individual runs as it might converge to only one of the multiple available robust optima in the different test functions. Nevertheless, it has the advantage of a good approximation of this local optimum, so, if it finds the correct optimum, this one is found with high precision and therefore we observe a better behavior of \sopt in the lower quantiles.
>
> In contrast to this, RES reasons not only about the robust optimum $f^\star$, but also about the argmax function $\mathbf{h}(\mathbf{x})$ and the corresponding maximizing function $g(\mathbf{x})$. In consequence, it has to learn about these multiple measures and its exploration behavior is (indirectly) reinforced. After a sufficient number of iterations, it is capable of a "rough" approximation of these measures, including the robust optimum $f^\star$, yielding a narrower distribution of regrets for each iteration than \sopt. As approximation quality improves, also the median values of this distributions become better than the ones by StableOpt, as the global robust optimum is better approximated.
>
> A natural approach to circumvent the risk of too small exploration of StableOpt is via fixing the hyperparameters in the more theoretical case that the model is known (we do this in the within-model comparison) or to perform a costly Bayesian treatment of the hyperparameters, that we perform in the robust robot pushing problem. In both cases, the difference of the median values is not as distinctive as in the experiments where the model hyperparameters are learned but remains if favor of our approach which seems to be more sample-efficient, but the behavior in the lower quantiles vanishes.
>
> We hope to have addressed your questions sufficiently and would be happy to further discuss the intuitions that we described. We propose to add a short version of them also to the manuscript. Nevertheless we have to ask if this is indeed the "Main Weakness b)" you mentioned in Q7.
>
> Additionally, we will try out your suggestion to collapse the results in figure 3 and decide on a change of the figure depending on the outcome, e.g., if the different lines of the ablation study on the number of samples $C$ are still visible. Furthermore, we are going to improve the paper, especially the technical part, until the end of rebuttal phase and for the camera-ready version.
>
> We hope to have sufficiently addressed your concerns and would be grateful for further comments from your side.

---

### Official Review · Reviewer_T2Hr · 2024-03-22

**Q2-1 Originality-Novelty:** 3
**Q2-2 Correctness-Technical Quality:** 2
**Q2-5 Clarity Of Writing:** 3

**Q1 Summary And Contributions:**

The motivation of this article is to meet the dual requirements of high sampling efficiency and finding robust solutions when using Bayesian optimization (BO) in engineering applications. The article proposes a method based on an efficient information acquisition function called Robust Entropy Search (RES), and demonstrates the advantages of this acquisition function through experiments on synthetic and real-world data. The results show that RES can reliably find robust optimal solutions, surpassing existing state-of-the-art algorithms.
The main contributions include defining adversarial robust optimization problems and proposing the RES method, which is a sample efficient information theory acquisition function.
A method was proposed to integrate adversarial robustness conditions into the probability distribution of obtaining the function core.
A rigorous empirical evaluation was conducted on the proposed method through the synthesis of data and practical scenarios in the fields of robotics and engineering.

**Q2-3 Extent To Which Claims Are Supported By Evidence:**

2: Fair: the main claims are somewhat supported by evidence (but the experimental evaluation may be weak, or does not match entirely with the claims, important baselines may be missing, proofs contain important ideas but lack rigor, algorithmic details are only discussed superficially, references are imprecise, assumptions are not sufficiently motivated or explicated, etc.).

**Q2-4 Reproducibility:**

3: Good: key resources (e.g. proofs, code, data) are available and key details (e.g. proofs, experimental setup) are sufficiently well-described for competent researchers to confidently reproduce the main results.

**Q3 Main Strengths:**

This work proposes a new Bayesian optimization method - Robust Entropy Search (RES), which is the first sample efficient information theory acquisition function for adversarial robust optimization problems, with high originality and novelty.
The article demonstrates the effectiveness of the RES method on synthetic and real-world datasets through theoretical analysis and experimental verification, demonstrating its technical quality. Especially in application examples in the fields of engineering and robotics, RES has shown superior performance. It provides sufficient evidence support for the effectiveness of the RES method through comparison with other advanced algorithms and experimental results on multiple benchmark functions.

**Q4 Main Weakness:**

Although RES performs well on specific problems, the article does not fully discuss the generalization ability and applicability of this method on different types of problems, it also does not discuss its calculation cost in detail, especially its performance on large-scale problems. The article does not provide in-depth analysis or guidance on hyperparameter selection and adjustment in RES methods, which may affect performance and usability in practical applications.

**Q5 Detailed Comments To The Authors:**

The introduction of robust entropy search (RES) as an innovative method in the field of Bayesian optimization undoubtedly provides a new perspective for solving adversarial robust optimization problems. However, in order to further enhance the influence and practicality of the work, I suggest the following improvement measures: although your experimental results are convincing, there is still a lack of discussion on generalization ability. I encourage you to test the RES method on more different types of optimization problems and report these experimental results in the article, which will help demonstrate the applicability and potential limitations of your method In terms of cost calculation, I noticed that your work did not discuss in detail the performance of the RES method in dealing with large-scale problems. To make your research more comprehensive, it is recommended to provide quantitative analysis on computational efficiency, including comparison of running time with other algorithms and scaling behavior on problems of different scales In addition, although the writing structure of the article is clear, there is still room for improvement in certain parts, especially in the detailed description of methodology.

**Q9 Complying With Reviewing Instructions:**

Yes

---

> ### Author Rebuttal · Authors · 2024-04-04
>
> Thank your for your helpful comments.
> Overall we understand from your review that there are two major concerns from your side: generalization capability and scalability of our approach.
>
> Let us begin with the scalability by giving additional results on calculation time. To run our approach, we have to optimize our derived acquisition function, just like other acquisition functions for Bayesian Optimization. The main cost of each evaluation of the acquisition function at a specific value of ${\mathbf{z} = (\mathbf{x}, \boldsymbol{\theta})}$ is governed by equation~(6) (Step 2 in 4.2.2: Creating a predictive distribution for a new location $\mathbf{z}$). For each function sample $f_c$, we have to find the value of the argmax function $\mathbf{h}_c(\mathbf{x})$ and its value $g_c(\mathbf{x}, \mathbf{h}_c(\mathbf{x}))$. In the case of uncontrollable variables, this inner optimization is performed using $N_r$ restarts of L-BFGS-B algorithm, each  iteration of it being of maximum order $\mathcal{O}(d_u)$ [1]. Each of the single evaluations of the inner optimization objective, e.g. the evaluation of the function sample $f_c$ goes with the cost of a matrix-vector multiplication of order $\mathcal{O}(M^2)$, where $M$ is the number of Fourier feature functions. In the case of discrete uncontrollable parameters, this step simplifies to a single prediction by the function sample $f_c$ for all $N_u$ uncontrollable parameters at the given $\mathbf{x}$, being still of order $\mathcal{O}(M^2)$ and the discrete maximization step of order $\mathcal{O}(N_u)$. Given the optima, we still have to marginalize over the function values $\mathbf{f}$ to create a prediction, which is of order $\mathcal{O}(N^3)$, where $N$ is the number of data points acquired so far.
>
> Putting these results together, we find that the runtime of our algorithm is governed by two competing effects: the number of data points acquired so far, and the effort for the inner optimization procedure. Assuming the inner optimization procedure runs for $N_i$ iterations, we find the runtime of our acquisition function scales by $O_{\text{cont.}} = C \cdot(\mathcal{O}(N^3) + N_r \cdot N_i \cdot (\mathcal{O}(d_u) + \mathcal{O}(M^2)))$ in the continuous case and $O_{\text{disc.}} = C \cdot (\mathcal{O}(N^3) + \mathcal{O}(M^2) + \mathcal{O}(N_u))$ in the discrete case.
>
> Overall, our implementation does not concentrate on computational efficiency, which can also be seen from the runtime experiments in the appendix. Nevertheless, its practical runtime is similar or better than the non-robust Knowledge Gradient acquisition function.
> Furthermore, we like to add that all common Bayesian Optimization approaches are not designed to large-scale problems, neither in the number of data points (as the acquisition functions make use of the Gaussian Process prediction, being of order $\mathcal{O}(N^3)$) neither in the number of dimensions, as enough data points are required to approximate the objective function, thus falling under the curse of dimensionality.
> Anyway, we will be happy to add the theoretical scalability results to our paper, thank you for mentioning this issue.
>
> Regarding the generalization capability of our approach, we like to state that we tried to cover the following variations in our experiments: from smooth (e.g., the Branin function) to wiggly (e.g., the Eggholder function), from completely continuous (e.g., the within-model comparison) over mixed problems (e.g., the Synthetic Polynomial) to fully discrete problems (e.g., the Hartmann 3d function), varying the dimensionality from 2 dimensions (e.g. the Branin function) to 5 (e.g. the Robot Pushing Problem). Nevertheless, we could run another more high-dimensional problem. Therefore, we propose to use the Hartmann-6 function, a 6-dimensional synthetic problem. We will provide the results as soon as they are ready and include them in CRC.
>
> Thank you as well for expressing your concerns about the presentation, especially in the method part. As already mentioned to reviewer Q4hk, we are going to improve writing and presentation until the end of rebuttal phase and afterward for the camera-ready version.
> Regarding the reproducibility issue you mentioned in Q7, we like to emphasize that our approach and the experimental setup is not only thoroughly described in the paper but that the code is available in the supplementary material and will be published at time of acceptance.
>
> We hope to have sufficiently addressed your concerns about the computational costs of our approach and would be grateful for further comments from your side.
>
> [1] Ciyou Zhu, Richard H. Byrd, Peihuang Lu, and Jorge Nocedal. Algorithm 778: L-bfgs-b: Fortran subroutines for large-scale bound-constrained optimization. ACM Transactions on Mathematical Software, 23(4):550–560, 1997

---

### Official Review · Reviewer_Q4hk · 2024-03-22

**Q2-1 Originality-Novelty:** 2
**Q2-2 Correctness-Technical Quality:** 2
**Q2-5 Clarity Of Writing:** 1

**Q10 Ethical Concerns:**

No.

**Q1 Summary And Contributions:**

This work proposes a robust entropy search acquisition function for safe Bayesian optimization.  In particular, the objective of interest takes two inputs: controllable and uncontrollable parameters, where controllable parameters are always adjustable and uncontrollable parameters are adjustable during an optimization process but cannot adjust at application time.  By considering a worst-case scenario, the robust entropy search has been proposed.  Finally the authors demonstrate experimental results on several optimization benchmarks including synthetic and real-world problems.

**Q2-3 Extent To Which Claims Are Supported By Evidence:**

2: Fair: the main claims are somewhat supported by evidence (but the experimental evaluation may be weak, or does not match entirely with the claims, important baselines may be missing, proofs contain important ideas but lack rigor, algorithmic details are only discussed superficially, references are imprecise, assumptions are not sufficiently motivated or explicated, etc.).

**Q2-4 Reproducibility:**

2: Fair: key resources (e.g. proofs, code, data) are unavailable but key details (e.g. proof sketches, experimental setup) are sufficiently well-described for an expert to confidently reproduce the main results.

**Q3 Main Strengths:**

- Problem formulation with controllable and uncontrollable parameters is interesting.
- Real-world experiments are also intriguing.

**Q4 Main Weakness:**

- Writing and presentation of this work can be improved more.
- Technical novelty is unclear.
- Some related work is not discussed.

**Q5 Detailed Comments To The Authors:**

- I think that the authors use `\citep` and `\citet` carefully.
- The mathematical font used in this work is somewhat different from one of the original UAI template.  I think some package that changes a font might be imported.  It should be fixed.
- Writing and presentation can be improved more by proof-reading the manuscript thoroughly.
- In Section 4.2.1, why is $\mathbf{a}$ a random vector?  Isn't it a fitting vector?
- Why are random Fourier features needed for modeling $f_c$?
- Why didn't the authors report best robust regrets or best inference/immediate regrets?  I think that best immediate (instantaneous) regrets, also known as simple regrets, are generally used in Bayesian optimization.
- For Figure 1(b), which $\theta$ is used to draw it?

**Q9 Complying With Reviewing Instructions:**

Yes

---

> ### Author Rebuttal · Authors · 2024-04-04
>
> We thank the reviewer for the detailed feedback.
>
> To begin with, let us briefly summarize our main contributions, addressing the concerns about technical novelty. We present the first information-based acquisition function to treat adversarially robust Bayesian optimization and show its performance in synthetic and real-world experiments. Our results show that it is not only better than typical non-robust benchmarks, but also more sample-efficient than the state-of-the-art algorithms.
>
> For the further discussion, we will provide detailed answers to the individual feedback and comments in the following.
>
> **Topic 1: Style and Presentation**
> We will happily proofread the paper again thoroughly, addressing all of the upper points. We have already improved on the correct usage of the citation commands. Additionally, we checked the reason for the different mathfont and found it being due to the choice of the option "newtx" instead of "ptmx" mathfont when loading the actual UAI template. We have chosen this font because we think it is easier to read. Thank you for pointing all the issues out.
>
> **Topic 2: Gaussian Process Samples**
> Thank your for pointing out your questions. We are going to revise the paper to make the paragraph easier to understand. In short, the random Fourier features are required to gain an unbiased approximation of the covariance function. The weights $\mathbf{a}$ are a random vector, as we are drawing random samples from the distribution over functions of the GP.
>
> **Topic 3: Choice of Performance Metric**
> Unfortunately, best regrets may be misleading for the robust problem, which is neither the pure minimum nor maximum. Simple regret is defined as the best function value obtained so far. With some efforts, this could be adapted to the min max setting, assuming $f^\star_t$ being the reported optimum of an algorithm, so we gain e.g. $r_n = \min_{t \in [1, n]} \lbrace f^\star_t - \min_{\mathbf{x} \in \mathcal{X}} \max_{\boldsymbol{\theta} \in \Theta} f(\mathbf{x}, \mathbf{\theta}) \rbrace$. Unfortunately, using this measure will suppress the diverging behavior of the non-robust (minimization) algorithms. At the beginning of optimization, they might find an optimum $f^\star_t$ that is close to the min max point, so their best regret will be low. In the further course of optimization, they will find smaller minima, thus diverging from the optimum. Using the best case policy, e.g., the minimum difference of the reported function value and the overall optimum, the best case metric would not be able to reflect this behavior.
> Thank you again for this question, we included this issue in the discussion of the used metrics in chapter 5.
>
> **Topic 4: Figure 1(b)**
> Let us briefly explain the intuition of the figure. We show the maximizing function $g(\mathbf{x})$, given a function $f(\mathbf{x}, \boldsymbol{\theta})$. The value of $\boldsymbol{\theta}$ used in figure 1(b) is obtained by calculating $\arg \max_{\boldsymbol{\theta} \in \Theta} f(\mathbf{x}, \boldsymbol{\theta})$, so $\boldsymbol{\theta}$ is not fixed in the figure, but corresponds to the white line in figure 1(a).
>
> **Topic 5: Related Work**
> You mention that we did not include all relevant related work. Could you give a hint which work you mean so we can include it?
>
> **Closing remarks**
> Thank you for your comments.
> We will be happy to fix the writing and presentation by intensive proofreading until the end of rebuttal phase afterward for the camera-ready version.
> We would be grateful if you increased your rating at the end of the rebuttal. We are glad to address the technical inaccuracies that you brought up, but like to also point out that the key characteristics (limited novelty, experimental evaluation, inadequate reproducibility, ethical considerations) that would warrant a reject rating are not given and neither have been put forward in your review.
> We hope to have sufficiently addressed your concerns and would be grateful for further comments from your side.

---

### Meta-Review · Area_Chair_NXvU · 2024-04-18

This paper proposes a new method for Bayesian Optimization (BO) that is particularly useful for engineering applications. In BO, the goal is to find the optimal value of a function by iteratively making queries and learning from the results. However, in engineering, there is often a need to consider both optimality and robustness to unexpected changes. This paper addresses this challenge by introducing a novel information-based acquisition function called Robust Entropy Search (RES).

The core idea of RES is to consider two types of uncertainties: the uncertainty about the overall best function value and the uncertainty about the function's robustness to small changes in the controllable parameters. The authors demonstrate that RES achieves good performance on various synthetic and real-world benchmarks.

Strengths:

Introduces a novel information-based acquisition function (RES) for robust Bayesian Optimization (BO).
Achieves good performance on synthetic and real-world benchmarks.
Code is available and well-described, enhancing reproducibility.
Shows promise for addressing practical engineering applications of BO.

Weaknesses:

Clarity and presentation require improvement, particularly in Section 4.2.1.
Limited discussion on generalization capabilities to different problem types.
Lack of in-depth analysis of hyperparameter selection and its impact.
Theoretical aspects like scalability and regret bounds could be strengthened.
Some reviewers raised concerns about the novelty compared to existing work.

Decision:

A majority of reviewers vote for acceptance.